



# Anisotropic Scattering in Radio-Echo Sounding: Insights from Northeast Greenland

Tamara Annina Gerber[1], David A. Lilien[2,3], Niels F. Nymand[1], Daniel Steinhage[4], Olaf Eisen[4,5], and Dorthe Dahl-Jensen[1,2]

[1]Section for the Physics of Ice, Climate and Earth, Niels Bohr Institute, University of Copenhagen, Copenhagen, Denmark.
[2]Centre for Earth Observation Science, University of Manitoba, Winnipeg, MB, Canada.
[3]Department of Earth and Atmospheric Sciences, Indiana University, Bloomington, IN, USA
[4]Alfred Wegener Institute Helmholtz Centre for Polar and Marine Research, Bremerhaven, Germany
[5]Department of Geosciences, University of Bremen, Bremen, Germany.

**Correspondence:** Tamara Annina Gerber (tamara.gerber@nbi.ku.dk)

**Abstract.** Anisotropic scattering and birefringence-induced power extinction are two distinct mechanisms affecting the azimuthal power response in Radio Echo Sounding (RES) of ice sheets. While birefringence is directly related to the crystal orientation fabric (COF), anisotropic scattering can, in principle, have various origins. Although both mechanisms can appear separately, they often act jointly, complicating efforts to deduce the COF strength and orientation from RES signals. In this

study, we assess the relative importance of anisotropic scattering and birefringence using quad-polarized ground-based RES measurements collected in the Northeast Greenland Ice Stream (NEGIS). We employ curve-fitting techniques to analyze the synthesized full azimuthal response, revealing insights into the dominance and orientation of the two different mechanisms at various depths between 630 m and 2500 m. We find that anisotropic scattering clearly dominates the radar signal in most depths larger than 1000 m, while birefringence effects are only important at shallower depths and in the vicinity of the ice-

stream shear margins. We further find that the co-polarized power difference follows the ice-sheet stratigraphy with a notable transition in strength and/or direction at the Wisconsin-Holocene transition and in folded ice outside the ice stream, possibly indicating inverted stratigraphy in these folded units. We conclude that small-scale fluctuations in the horizontal COF eigenvalues is the most likely mechanism responsible for the anisotropic scattering observed in our survey area. Mapping the strength and orientation of scattering in quad-polarized measurements thus have the potential to provide independent estimates of the

COF orientation and distinguish ice units with different scattering properties, e.g. from different climatic periods.

## 1 Introduction

Radio echo sounding (RES) is widely used for studying ice sheets, glaciers, and ice caps by sending electromagnetic waves in the radio frequency range into the ice and measuring the strength of the reflected signals as a function of time. These radio



echoes contain valuable information about the subsurface structure and properties of ice masses, which can provide crucial insight into the past and present state of the surveyed ice bodies. An important observation in RES measurements is the azimuthal dependency of the return power, where the strength of the RES signal varies with the orientation of the polarization. Azimuthal power fluctuations have been observed in previous studies involving polarimetric measurements, here defined as RES measurements including multiple polarization directions. With co-polarized systems, where transmit and receive polar-

izations are parallel, multiple polarizations can be recorded when the radar system moves on a circular path or *turning circle* (Eisen et al., 2007; Matsuoka et al., 2012; Drews et al., 2012). For co-polarized airborne RES systems, where turning circles are difficult to implement, crossing flight paths yield two polarizations (Mojtabavi et al., 2019; Gerber et al., 2023). Another method involves the manual rotation of co-polarized and/or cross-polarized (perpendicular transmit and receive polarization) antennas in discrete increments on the spot in ground-based surveys (Fujita et al., 2006; Brisbourne et al., 2019; Jordan et al.,

2020; Young et al., 2021). The full azimuthal power response can also be synthesized from quad-polarimetric measurements, where transmit and receive polarizations are sequentially rotated by 90° resulting in two co-polarized and two cross-polarized modes (Ershadi et al., 2022). The azimuthal dependency of polarimetric radar return power is influenced by two distinct mechanisms; birefringence and anisotropic scattering (Fujita et al., 2006; Matsuoka et al., 2009). While both mechanisms can appear independently of each other, they often occur simultaneously to some degree.

Birefringence is directly related to the preferred alignment of ice crystals, commonly known as crystal orientation fabric (COF) or lattice preferred orientation. Single ice crystals are transversely isotropic about the crystal axis (c-axis), with higher dielectric permittivity along the c-axis than along the basal plane. This leads to bulk birefringence of polycrystalline ice with a preferred orientation of ice crystals. Propagating electromagnetic waves are decomposed into two orthogonally polarized waves, each polarized in the direction of one principal horizontal COF axes (Hargreaves, 1977). For a vertically propagating

wave, which is the standard operation in glaciological applications, and assuming that one eigenvector is vertical, the polarizations of the two waves are in the horizontal plane, and their wave speeds depend on horizontal anisotropy. For a transmit polarization which is not aligned with the principal COF axis these two waves can appear as *double reflections* in radargrams, i.e. two reflections caused by the same physical interface but arriving at the receiver with a time delay due to different wave speeds (Nymand et al., 2024). Additionally, interference between the two waves leads to power extinction nodes when the phases are shifted by half a wavelength upon reception (Fujita et al., 2006). In radargrams, these appear with a vertical spac-

ing as a function of the strength of horizontal anisotropy and radar frequency (Young et al., 2021; Gerber et al., 2023). This birefringence-induced power modulation appears with an azimuthal periodicity of 90° in both co-polarized and cross-polarized measurements. In the absence of anisotropic scattering, co-polarized extinction (CoPE) nodes appear at an azimuth of 45° from the principal COF axis, while cross-polarized extinction (XPE) aligns with COF axes (see e.g. Fig. 5 in Fujita et al., 2006, for an illustrative overview). The effect remains present, with 90° azimuthal periodicity, even if none of the principal directions of

the COF are vertical (Rathmann et al., 2022).

Anisotropic scattering refers to the directional dependence of the scattering properties of ice, which leads to variations in the intensity of the received signal depending on antenna orientation. Scattering arises from mechanisms related to the inherent micro-structural and textural properties of ice (volume scattering), or from reflections off internal reflection horizons (surface




scattering) (e.g. Langley et al., 2009; Drews et al., 2012). Volume scattering is caused by small-scale inhomogeneities in the physical properties affecting the overall return power such as air bubbles, dust particles or impurities. Anisotropy can arise if these scattering sources are rotationally asymmetric, for example when air bubbles or other inclusions are elongated, or due to small-scale COF fluctuations (Drews et al., 2012). Anisotropic volume scattering is also observed in the optical range of the electromagnetic spectrum, where waves are more sensitive to small inhomogeneities (e.g. Rongen et al., 2020; Abbasi et al., 2024). Surface scattering occurs at reflection horizons which are typically associated with volcanic eruptions or climatic transitions (Paren and Robin, 1975; Fujita et al., 1999; Hempel et al., 2000; Eisen et al., 2006), material boundaries such as water, air or sediment inclusions (Robin et al., 1969; Paren and Robin, 1975), or, in some cases, at abrupt COF changes (Eisen et al., 2007). Anisotropic reflections appear when these interfaces are of anisotropic character, for example, due to directional roughness (e.g. van der Veen et al., 2009; Cooper et al., 2019; Eisen et al., 2020), or COF transitions involving horizontal anisotropy (Eisen et al., 2007). Regardless of its physical origin, anisotropic scattering can be distinguished from pure birefringence effects by its azimuthal periodicity of 180° (as opposed to the 90° birefringence periodicity) in co-polarized return power (Fujita et al., 2006).

Most previous polarimetric studies aimed to determine the horizontal component and orientation of the electromagnetic anisotropy, which can be related to the horizontal COF anisotropy. The COF type and strength varies with depth, location, and ice formation conditions (Gow and Williamson, 1976; Faria et al., 2014b; Llorens et al., 2022). In-situ measurements of COF, e. g. along ice cores, are important to understand and simulate the effect of the resulting mechanical anisotropy on ice deformation and flow dynamics (e.g. Duval et al., 1983; Budd and Jacka, 1989; Lilien et al., 2021; Llorens et al., 2022). The COF anisotropy can be derived through analyzing CoPE and XPE (Young et al., 2021; Gerber et al., 2023) and/or the coherence phase (Dall, 2010b; Jordan et al., 2019; Ershadi et al., 2022; Jordan et al., 2022). CoPE nodes in particular can be used to estimate the horizontal COF anisotropy across large areas, since they can also appear in single-polarized radar measurements (Young et al., 2021; Gerber et al., 2023), which is the most commonly used RES type for large-scale airborne surveys of ice sheets. The presence of anisotropic scattering, however, can complicate these analyses by altering the vertical elongation and azimuthal distance between CoPE nodes and the angular width of dipole nodes (DN) in coherence phase (Fujita et al., 2006; Ershadi et al., 2022). Understanding the relative importance and potential origin of anisotropic scattering is therefore useful when attempting to derive the strength and orientation of COF anisotropy from birefringence signatures in radargrams.

In this study, we use RES data recorded by a ground-based quad-polarized ultra-wideband radar to quantify the relative importance of anisotropic scattering and birefringence effects in a highly dynamic area of the Greenland Ice Sheet (GrIS). Our study site (see Fig. 1) is located in the onset region of the Northeast Greenland Ice Stream (NEGIS) in the vicinity of the drill site from the East Greenland ice-core project (EastGRIP). The NEGIS region is of particular interest due to fast ice flow (reaching 55 ma$^{-1}$ at the drill site; Hvidberg et al., 2020) where deformation-induced anisotropic scattering mechanisms can be expected, and the COF is known to be highly anisotropic in the ice-stream center (Stoll, 2019; Zeising et al., 2023; Gerber et al., 2023). In our analysis, we synthesize the full azimuthal response from quad-pol measurements, which we validate by comparison to the azimuthal response from a radargram recorded in a turning circle and modeling with the COF record from the EastGRIP ice core. We then determine the relative amplitude and orientation of 90° and 180° periodical power fluctuations



with curve-fitting methods for different depth intervals along radar lines at a spacing of 5 km. We find that anisotropic scattering is dominant over most of the survey area and particularly strong inside the ice stream with orientation near-perpendicular to ice-flow direction. Furthermore, we observe that the strength and orientation of anisotropic scattering is related to the ice-sheet stratigraphy and shows an abrupt reversal in directionality at the 11.4 ka isochrone, corresponding to the transition from Holocene into Wisconsin ice, at a depth of roughly 1300 m outside the shear margins. We discuss potential mechanisms for anisotropic scattering and conclude that small-scale vertical variations in the COF are the most likely origin for the observed anisotropic scattering patterns.

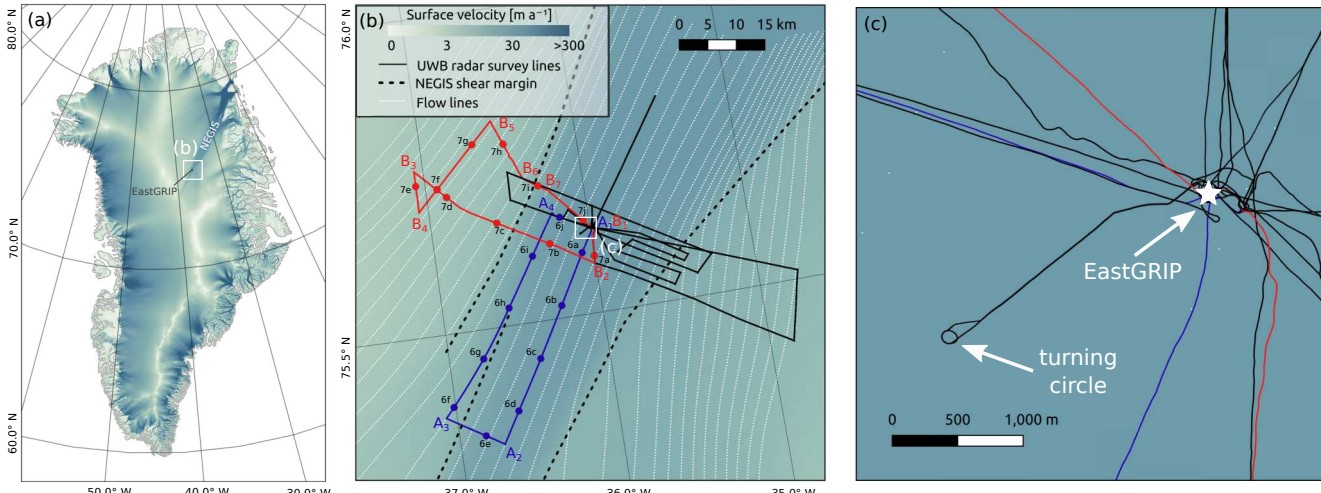

**Figure 1. (a)** Overview of the radar survey near the EastGRIP ice core located in the Northeast Greenland Ice Stream (NEGIS), indicated by the enhanced surface flow velocities (Joughin et al., 2018). **(b)** Polarimetric RES lines are shown as black solid lines with two example radargrams (profiles A and B shown in Fig. 9) highlighted in blue and red respectively. $A_1$, $A_2$, etc. indicate reference points of changing driving direction in Fig. 2 and Fig. 9, and dots along profile A (6a–6j) and B (7a–7j) mark analysis points in Fig. 6 and Fig. 7. Thick dashed lines indicate the position of the shear margins and white lines indicate the streamlines derived from surface velocities. **(c)** Location of the turning circle near the EastGRIP drill site.

## 2 Radio-echo sounding data and isochrone dating

The RES data used in this study were recorded in June/July 2022 with a quad-polarized ground-based system. The radar operated at a center frequency of 330 MHz with 300 MHz bandwidth and a chirp length of 10 μs. Eight channels from the digital system were divided to ten amplifiers with peak transmit power of 150–250 W per amplifier, feeding a $10 \times 10$ element array. This setup formed a 2.7 m $\times$ 2.7 m antenna array resting on a balloon (see Yan et al., 2020) and was dragged over the snow surface at an average speed of $\sim$3 m s$^{-1}$. A high-power switch coordinated the polarization of transmit and receive



channels for quad-polarized recording (HH, HV, VV, and VH, where V denotes along-track polarization). The receivers were blocked during the 10 µs transmit time, corresponding to an approximate depth of 630 m for the first recorded returns (described

in detail by Yan et al., 2020).

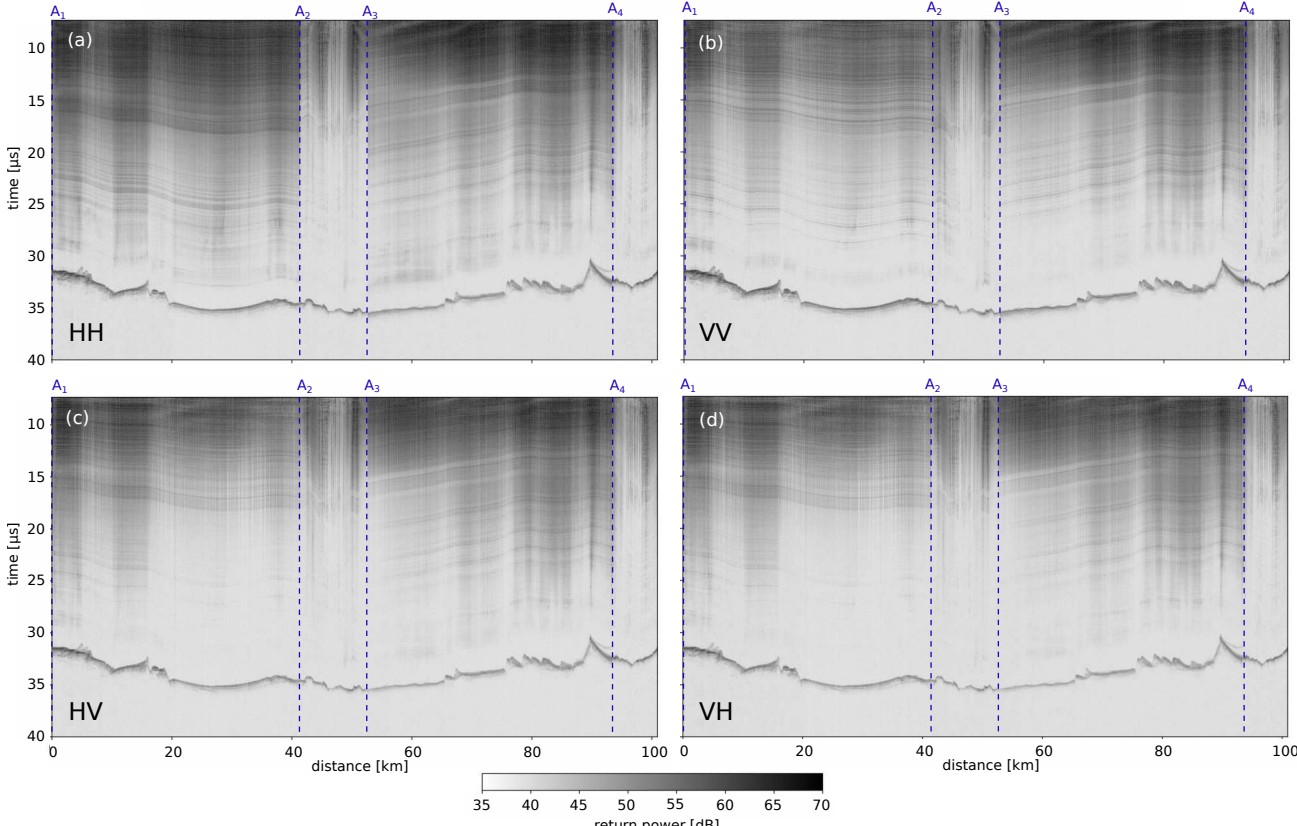

**Figure 2.** Example radargram (profile A in Fig. 1) recorded with the quad-polarized radar. Co-polarized profiles are shown in panel (**a**) (HH—cross-track) and (**b**) (VV—along-track). Panels (**c**) and (**d**) show cross-polarized profiles (HV and VH, where the first and last letters indicate transmit and receive polarization, respectively.)

The radar survey was conducted along the lines shown in Fig. 1, which are mostly aligned parallel/perpendicular to the general southwest to northeast surface ice-flow direction and cover a distance of approximately 450 km, spanning over the entire width of the ice stream. Figure 2 shows an example radargram (profile A in Fig. 1) in the four polarization modes HH, HV, VV, and VH. Additionally, a circular radargram with an approximate radius of 50 m was recorded near the EastGRIP camp

for testing purpose (Fig. 1c). Data processing followed a standard procedure including coherent integration, pulse compression, incoherent integration, channel integration and interpolation to a consistent grid (Nymand et al., 2024).





Isochrones were traced manually for profiles A and B (Fig. 9), and dated following Gerber et al. (2021). In doing so, the travel times were converted to depth using a velocity profile inferred from the relative dielectric permittivity obtained from Dielectric Profiling (DEP) of the EastGRIP core (Mojtabavi et al., 2022). The DEP complex-valued, ordinary relative dielectric

permittivity, $\varepsilon$ was measured at 250 kHz at a resolution of 5 mm, starting at a depth of 13.7 m. We used a smoothed version of the real part of the dielectric record, $\varepsilon'(z)$, to estimate the velocity profile in the upper 183 m by extrapolating to a surface value of $\varepsilon'_s = 1.55$. Below the transition into pure ice at a depth of 183 m we assume a constant relative permittivity in ice of $\varepsilon'_i = 3.15$. The velocity profile is then obtained with

$$c(z) = \frac{c_0}{\sqrt{\varepsilon'(z)}}, \tag{1}$$

where $\varepsilon'(z)$ is the relative dielectric permittivity and $c_0$ is the speed of light. The age uncertainties generally increase with depth due to the timescale maximum counting error, and because the existing timescale does not extend to the deepest layer traced here (see Gerber et al., 2021, for discussion of uncertainties). The deepest traced isochrone (74.7 ka, see Fig. 9) has an additional uncertainty from the layer tracing across the shear margin in profile B, particularly since the bottom part from 20 km and onward is heavily folded, causing ambiguity in matching the deepest isochrone inside and outside the shear margin.

## 3 Azimuthal response - model and observations at EastGRIP

The effects of anisotropic scattering and birefringence can be distinguished by their periodicity of co-polarized power anomalies. The full angular response was measured by the radar in a turning circle with approximate diameter of 50 m, but can also be synthesized from a single quad-pol measurement (Ershadi et al., 2022). For assessing the relation between COF, birefringence and anisotropic scattering, we compare power anomalies from a wave-propagation model using the COF record from 130 the nearby EastGRIP ice core with the turning circle and the synthesized azimuthal response from quad-pol measurements.

### 3.1 Simulating azimuthal response with ice-core COF

We use the matrix-based layer model for two-way radar wave propagation by Fujita et al. (2006) to simulate the theoretical azimuthal response at EastGRIP. For nadir-propagating waves, as is the case here, a $2 \times 2$ matrix model is sufficient, but $4 \times 4$ matrices are needed otherwise (Rathmann et al., 2022). Our notation and the model description closely follows Fujita et al.
(2006) - readers familiar with that work are invited to skip the following paragraph.

The signal received at the surface is described by

$$\mathbf{E_R} = \frac{\exp(jk_0 z)}{4\pi z} \prod_{i=1}^{N} [\mathbf{R}(\theta_{N+1-i})\mathbf{T}_{N+1-i}\mathbf{R}(-\theta_{N+1-i})]\mathbf{E}'_i, \tag{2}$$

where $j^2 = -1$, $\mathbf{R}$ is the rotation matrix, $\theta$ is the polarization azimuth angle, $k_0$ is the wave number in a vacuum, $z$ is depth below surface, $\mathbf{T}$ is the transmission matrix and $\mathbf{E}'_i$ is the electrical field scattered off from scattering boundaries in layer $i$

$$\mathbf{E}'_i = \mathbf{R}(\theta_i)\mathbf{S}_i\mathbf{R}(\theta_i)\mathbf{E}_i, \tag{3}$$



with $\mathbf{S}_i$ being the scattering matrix and $\mathbf{E}_i$ being the electrical field incident to scattering boundary $i$

$$\mathbf{E}_i = \frac{\exp(\mathrm{j}k_0 z)}{4\pi z} \prod_{i=1}^{N} [\mathbf{R}(\theta_i)\mathbf{T}_i\mathbf{R}(-\theta_i)]\mathbf{E}_{\mathrm{T}}. \tag{4}$$

$\mathbf{E}_{\mathrm{T}}$ is the transmitted electrical field. The scattering matrix of layer $i$, is defined as

$$\mathbf{S}_i = \begin{pmatrix} S_{x,i} & 0 \\ 0 & S_{y,i} \end{pmatrix}, \tag{5}$$

where $S_{x,i} = S_{y,i}$ for the case of isotropic scattering. The transmission matrix at layer $i$ is defined as

$$\mathbf{T}_i = \begin{pmatrix} T_{x,i} & 0 \\ 0 & T_{y,i} \end{pmatrix}, \tag{6}$$

with

$$\begin{aligned} T_{x,i} &= \exp(-\mathrm{j}k_0\Delta z_i + \mathrm{j}k_x\Delta z_i), \\ T_{y,i} &= \exp(-\mathrm{j}k_0\Delta z_i + \mathrm{j}k_y\Delta z_i), \end{aligned} \tag{7}$$

where $\Delta z_i$ is the thickness of layer $i$, and

$$\begin{aligned} k_x &= \sqrt{\varepsilon_0\mu_0\varepsilon_x'\omega^2 + \mathrm{j}\mu_0\sigma\omega}, \\ k_y &= \sqrt{\varepsilon_0\mu_0\varepsilon_y'\omega^2 + \mathrm{j}\mu_0\sigma\omega}, \end{aligned} \tag{8}$$

with vacuum dielectric permittivity $\varepsilon_0 = 8.85{\times}10^{-12}$ A s V$^{-1}$m$^{-1}$, the vacuum magnetic permeability $\mu_0 = 4\pi{\times}10^{-7}$V s A$^{-1}$m$^{-1}$, $\varepsilon_x'$ and $\varepsilon_y'$ being the directional relative permittivities in $x$- and $y$-direction, the angular frequency $\omega = 2\pi f_c$ with center frequency $f_c = 330$ MHz, and the electrical conductivity $\sigma$ is taken to be $2{\times}10^{-5}$ S m$^{-1}$.

Analogous to Fujita et al. (2006), we simulate the individual response of birefringence and anisotropic scattering, as well
as the combined effect. For cases with birefringence we use the EastGRIP COF record by Zeising et al. (2023) (Fig. 3a) to calculate the components of the transmission matrix $\mathbf{T}$. The COF has been analyzed in discrete samples at intervals of 10–15 m using an automated fabric analyzer and is statistically described in terms of eigenvalues, with its two horizontal components $\lambda_x$ and $\lambda_y$, and $\lambda_z$ being vertical (Fig. 3a). We define the model $x$-direction as being parallel to the ice-flow direction at EastGRIP, and assume $\lambda_x$ is flow-parallel following observations by Westhoff et al. (2021). The directional relative permittivity profiles
$\varepsilon_x'$ and $\varepsilon_y'$ are calculated from the two horizontal grain-size weighted eigenvalues $\lambda_x$ and $\lambda_y$

$$\varepsilon_{x,y}'(z) = \varepsilon_{\perp}' + \Delta\varepsilon'\lambda_{x,y}, \tag{9}$$

where $\varepsilon_{\perp}' = 3.15$ is the relative permittivity component perpendicular to the c-axis, and $\Delta\varepsilon'$ is assumed to be 0.034 (Matsuoka et al., 1997). We use a model with a regular layer thickness of 1 m, ranging from 111 m to 1700 m and interpolated the COF dataset linearly between measurements to match the vertical resolution of the model.





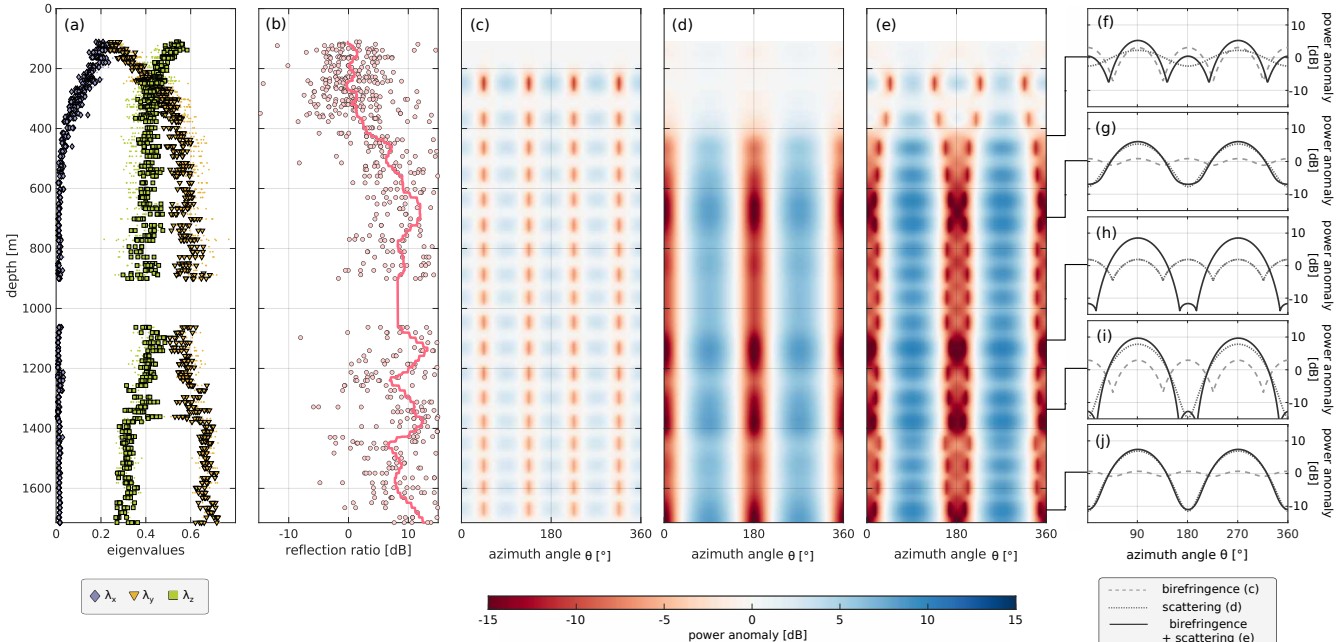

**Figure 3.** Simulated radar response with 330 MHz center frequency and ice-core derived scattering and birefringence properties: Panel (**a**) shows the EastGRIP COF eigenvalues from Zeising et al. (2023), where purple correspond to $\lambda_1$, orange to $\lambda_2$ and green to $\lambda_3$. $\lambda_x$ is approximately parallel to surface flow direction at EastGRIP. Dots of the corresponding colors are the pure eigenvalues, while squares, triangles and diamonds are the eigenvalues weighted to grain size. Panel (**b**) shows the reflection ratio derived form Eq. (12) (dots) and smoothed over 50 m (line), assuming that the eigenvalue variation with depth is its sole contribution. Panels (**c–e**) show the co-polarized power anomalies (dP$_{HH}$) of the radar response from birefringence, COF-induced anisotropic scattering and the combined effect simulated with the matrix-based model by Fujita et al. (2006). y-axes correspond to depth along the ice core and x-axes indicate polarization azimuth with $\theta = 0$ corresponding to model $x$-direction (i.e. $\lambda_x$, assumed to be flow-parallel). Panels (**f–j**) show the power anomalies as a function of azimuth for selected depths, indicated by lines connected to panel (e).

Our modeling assumes that the sole origin for anisotropic scattering is COF variations with depth, so we can estimate the components of the scattering matrix $\mathbf{S}_i$ from the impedance-derived scattering coefficient following Fujita et al. (2000)

$$S_{x,i} = \frac{Z_{x,i+1} - Z_{x,i}}{Z_{x,i+1} + Z_{x,i}} \text{ and } S_{y,i} = \frac{Z_{y,i+1} - Z_{y,i}}{Z_{y,i+1} + Z_{y,i}} \tag{10}$$

with impedance in layer $i$ defined as

$$Z_{x,i} = \frac{\mathrm{j}\mu_0\omega}{k_{x,i}} \text{ and } Z_{y,i} = \frac{\mathrm{j}\mu_0\omega}{k_{y,i}}. \tag{11}$$

The anisotropic reflection ratio in dB is then

$$r_i[\mathrm{dB}] = 10\log_{10}\left(\frac{S_{y,i}}{S_{x,i}}\right). \tag{12}$$



We smoothed the scattering coefficients with a moving average filter with window length of 50 m. The transmission and reflection matrices were kept isotropic for the pure scattering ($T_x = T_y$) and pure birefringence ($S_x = S_y$) scenarios, respectively.

The co- and cross-polarized power anomalies dP$_{HH}$ and dP$_{HV}$ are defined as (Dall, 2010a; Jordan et al., 2019; Ershadi et al., 2022)

$$\mathrm{dP}_{xx}(\theta, z) = 20 \log_{10}\left( \frac{|M_P^{xx}(\theta, z)|}{\frac{1}{n}\sum_{b=1}^{n} |M_P^{xx}(\theta_b, z)|} \right), \tag{13}$$

where $M_P^{xx}$ corresponds to the HH and HV component of the propagation matrix

$$\mathbf{M}_P = \begin{pmatrix} M_P^{HH} & M_P^{HV} \\ M_P^{HV} & M_P^{VV} \end{pmatrix}, \tag{14}$$

and $n$ is the number of angular increments of azimuth $\theta$ (Ershadi et al., 2022). Here, we introduce the term *propagation matrix* for $\mathbf{M}_P$ instead of the *scattering matrix* used in the definition by Ershadi et al. (2022) in order to avoid confusion with the scattering matrix used above in the sense of the definition by Fujita et al. (2006). The propagation matrix combines the propagation in the subsurface (see Appendix A for full description)

$$\mathbf{E}_R = \mathbf{M}_P \mathbf{E}_T, \tag{15}$$

where $\mathbf{E}_R$ and $\mathbf{E}_T$ are the received and transmitted electrical fields.

The co-polarized power anomalies dP$_{HH}$ for the pure birefringence, pure anisotropic scattering and combined effects are shown in panel c–f in Fig. 3, where an azimuth of $\theta = 0°$ corresponds to the model $x$-direction, i.e. the approximate surface flow direction at EastGRIP in $\sim33°$ clockwise from True North. The vertical spacing between birefringence nodes is approximately 120 m in the top 400 m and decreases to 85 m at greater depths. The amplitude of the largest power extinction nodes for pure birefringence is $-10$ dB, while the power maxima in panel c is 4 dB. The effect of anisotropic scattering becomes most notable at a depth of 400 m and below, with maximum return power of up to 8 dB at 90° and 270° and power minima of $-15$ dB at a 90° angular offset (Fig. 3d). Anisotropic scattering clearly dominates the azimuthal power distribution in the combined model (panel e) at depths larger than 400 m. Local maxima in anisotropic scattering occur at depths of 680 m (panel g), 1140 m (panel h), 1366 m (panel i), and 1692 m (panel j).

### 3.2 Comparison of model and RES

Now we compare the ice-core based model response of both birefringence and anisotropic scattering (Fig. 3c), with observations from RES data. For this comparison we used a reflection ratio of $0.5 \times r[\mathrm{dB}]$ in Fig. 3b to match the observed scattering amplitudes. The full azimuthal response can be synthesized from single quad-polarized measurements (Ershadi et al., 2022) with

$$\mathbf{M}_P(\theta + \gamma) = \mathbf{R}(\theta + \gamma)\mathbf{M}_P(\theta)\mathbf{R}'(\theta + \gamma), \tag{16}$$

where $\gamma$ is the angular offset between the initial radar orientation $\theta$ and the desired azimuth angle.





The complex co-polarized coherence for depth $n$ is defined as

$$C_{\mathrm{HHVV},n} = \frac{\sum_{b=n}^{n+N} M_{\mathrm{P},b}^{\mathrm{HH}} M_{\mathrm{P},b}^{*\mathrm{VV}}}{\sqrt{\sum_{b=n}^{n+N} |M_{\mathrm{P},b}^{\mathrm{HH}}|^2}\sqrt{\sum_{b=n}^{n+N} |M_{\mathrm{P},b}^{\mathrm{VV}}|^2}}, \tag{17}$$

where $N$ is the number of depth bins used for averaging and $*$ is the complex conjugate. Here, we average over a vertical depth of 50 m. The coherence phase difference then follows as

$$\phi_{\mathrm{HHVV}} = \arg(C_{\mathrm{HHVV}}), \tag{18}$$

and the normalized gradient of $\phi_{\mathrm{HHVV}}$ is

$$\psi_{\mathrm{HHVV}} = \frac{2c_0\sqrt{\varepsilon'}}{4\pi f_c \Delta\varepsilon'}\frac{\mathrm{d}\phi_{\mathrm{HHVV}}}{\mathrm{d}z}. \tag{19}$$

Figure 4 shows the azimuthal power anomalies (dP$_{\mathrm{HH}}$, dP$_{\mathrm{HV}}$), the coherence phase difference ($\Phi_{\mathrm{HHVV}}$) and its depth gradient ($\psi_{\mathrm{HHVV}}$) for the turning circle (panel a–d), synthesized (panel e–h) and modeled (panel i–l) data, respectively, whereby the

azimuth on the $x$-axes corresponds to clockwise angle from True North. The synthesized response was calculated from a single point at the beginning of the turning circle, where the driving direction was constant. This point was chosen to mitigate the impact of azimuthal discrepancies caused by the integration process, which smooths each polarization mode over a horizontal distance of approximately 3 m. The power anomalies and coherence phase difference/gradient of the turning circle and the synthesized response (panel a–h in Fig. 4) have been smoothed with a 2D Gaussian filter with standard deviation of 10 pixels.

The features in the turning circle and synthesized response match quite well, which gives confidence in using the synthesized response to evaluate azimuthal power fluctuations elsewhere. The general pattern in the radar data consistently shows highest co-polarized return power at an azimuth of roughly 135-140° and 315-320°. However, in the modeled response, this pattern is slightly shifted with respect to the observed data by ~10–15° and maximum co-polarized power appears at 125° and at 305°, respectively. The radar-derived amplitude of the dP$_{\mathrm{HH}}$ power anomaly increases at a depth of ~1400 m and is also slightly

higher at ~1700 m, both of which is also shown by the model. The latter additionally shows a higher dP$_{\mathrm{HH}}$ at 1150 m depth which is not confirmed by radar observations. Instead, depths shallower than 1300 m in the radar data are characterized by lower amplitudes and a pattern dominated by 90° periodicity (see below).

The observed and synthesized cross-polarized power anomalies (dP$_{\mathrm{HV}}$, see panel b, f) show a notable azimuth shift of the XPE at 1250 m depth towards smaller azimuth angles which is not shown in the model. The underlying cause of this is

presumably a rotation of COF eigenvectors not captured in the reconstructed orientation of the ice core (Zeising et al., 2023). However, the XPE azimuth below 1250 m is constant with depth and agrees well between the modeled and observed data for the depth span covered by the model, although here too, the azimuth between model and radar is shifted by approximately 10°.

The modeled coherence phase shows a large number of dipole nodes (DN) with an angular width of approximately 15–30° while individual nodes are hard to recognize in the turning circle and synthesized response, and angular widths are notably

smaller. The phase–depth gradient can, in principle, be used as indication of eigenvector orientation, with negative gradients indicating the orientation of the smaller horizontal eigenvalue (Jordan et al., 2019; Ershadi et al., 2022). While this can be





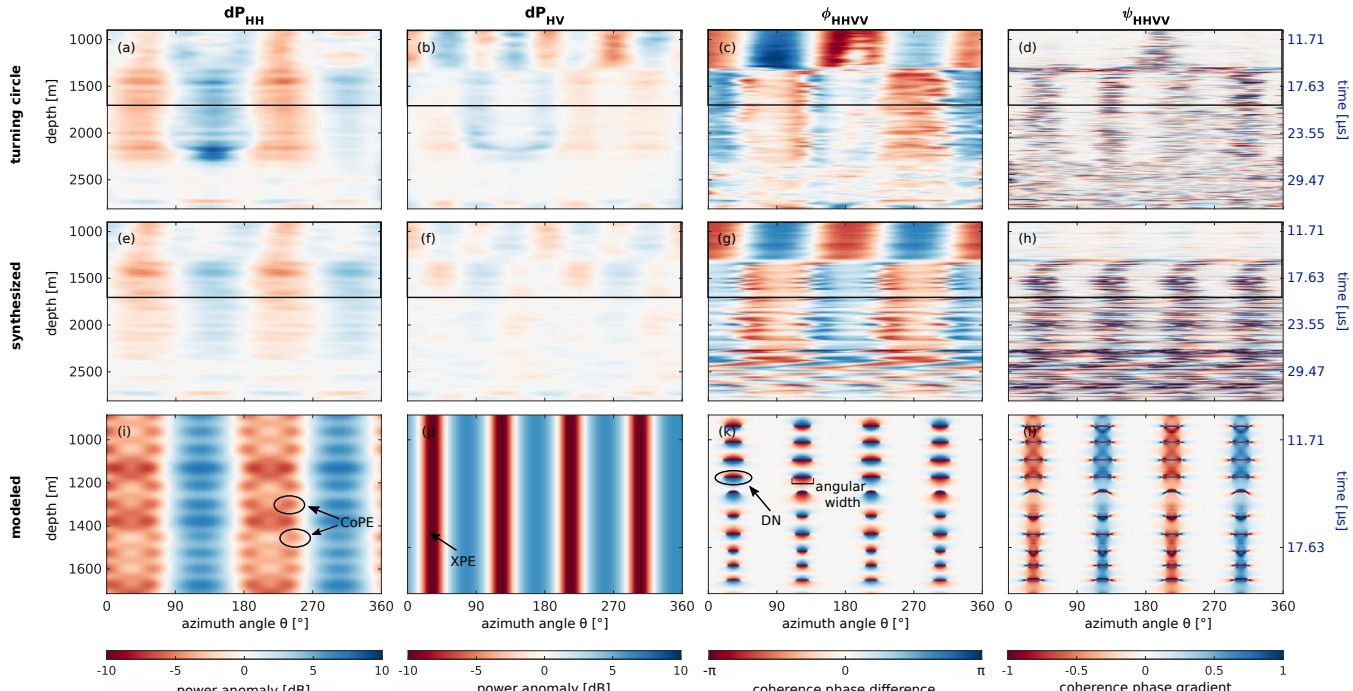

**Figure 4.** From left to right, the figure shows co-polarized power anomaly (dP$_{HH}$; panel (**a**), (**e**), (**i**)), cross-polarized power anomaly (dP$_{HV}$; panel (**b**), (**f**), (**j**)), coherence phase difference ($\phi_{HHVV}$; panel (**c**), (**g**), (**k**)), and the depth gradient of $\phi_{HHVV}$ ($\psi_{HHVV}$, panel (**d**), (**h**), (**l**)) versus polarization azimuth clockwise from True North. From top to bottom, the panels show the radar response from the circular radargram (panel (**a–d**)), synthesized from the quad-polarized measurements from the same radargram (panels (**e–h**)), and modeled with the COF record from the EastGRIP core (panels (**i–l**)). Notice the different depth range of the modeled result (indicated by the black frame in panel (a)–(h)) due to restrictions from available COF data. Co-polarized power extinction (CoPE) nodes, cross-polarized power extinction (XPE), coherence phase dipole nodes (DN) and their angular width are exampled in panels (**i–k**).

confirmed by the model, the phase gradient of the observed and synthesized data turn out to be too noisy to derive eigenvector orientations. However, the width of the zones with phase-gradient close to zero is similar between model and observations, indicating that the amount of anisotropic scattering between the two is comparable (Jordan et al., 2019). In the synthesized data, and partially also in the turning circle, the phase gradient above ~1250 m depth is considerably smaller than below, which might indicate weaker horizontal anisotropy, but is not confirmed by the model.

To quantitatively estimate the relative importance of anisotropic scattering (180° dP$_{HH}$-periodicity) and birefringence (90° dP$_{HH}$-periodicity), we fit the sum of 90°- and 180°-periodic sinusoidal signals to dP$_{HH}$ as a function of azimuth at five depth intervals (Fig. 5)

$$f(\theta) = A_{180}\sin(2\theta + \varphi_{180}) + A_{90}\sin(4\theta + \varphi_{90}), \tag{20}$$



where $\theta$ is the azimuth angle. The relative importance of anisotropic scattering and birefringence is given by the fitted amplitudes $A_{180}$ and $A_{90}$, respectively, which are displayed on top of each panel in Fig. 5, and scattering direction is given by the fitted phases $\varphi_{180}$ and $\varphi_{90}$. For all three datasets, anisotropic scattering becomes increasingly important with increasing depth, as the $\frac{A_{180}}{A_{90}}$ ratio increases for all depth increments except between panel g and j. Birefringence is only dominant in shallow
depths, notably in the 0–500 m interval of the model output (panel a), and the 500–1000 m interval of the turning circle (panel b). Amplitudes of anisotropic scattering are generally higher in the model, compared to radar observations.

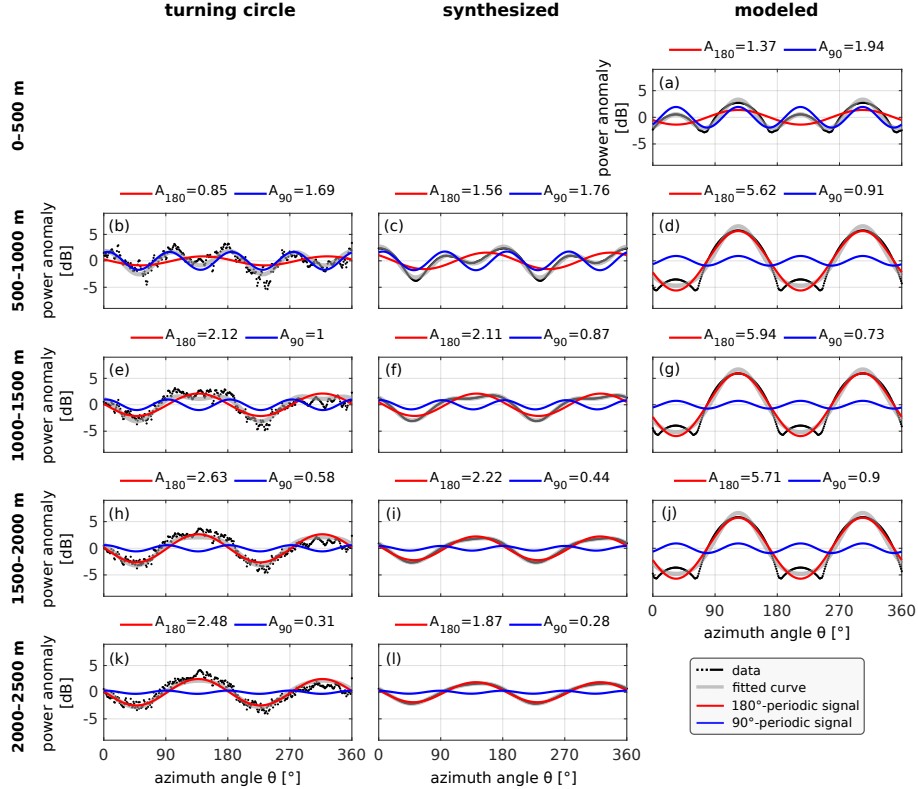

**Figure 5.** Depth-averaged power anomalies from turning circle (left), synthesized (middle), and modeled with the COF at EastGRIP (right) shown in black. For all depth intervals indicated on the left side of the figure, the sum of a 90° (blue) and 180° (red) periodic signal are fitted to the data points (black) using Eq. (20) and is shown in gray. The amplitudes of the corresponding 90°- and 180°-signal are displayed on top of each panel, indicating the relative importance of birefringence (90° periodicity) and anisotropic scattering (180° periodicity). The x-axes indicate azimuth as clockwise angle from True North in all panels.

## 4    Orientation and strength of scattering and birefringence across NEGIS

We now investigate the strength and orientation of anisotropic scattering on larger spatial scales by calculating the synthesized co- and cross-polarized power anomalies and coherence phase difference akin to panel e–g in Fig. 4 at intervals of 5 km along



the remaining radargrams. We use the same curve-fitting procedure as in Fig. 5 to determine the orientation and strength of anisotropic scattering and birefringence effects for different depths. The synthesized response is calculated from an average over 20 traces to improve signal-to-noise level, since the driving direction at the analysis points is reasonably constant (which was not the case in the turning circle) and the subsurface properties are not expected to change significantly over the corresponding distance (∼100 m). The curve-fitting was done at every 20 m depth on the synthesized angular response averaged over a 0.2 µs interval, or ∼16 m depth. Figures 6 and 7 show the power anomalies and coherence phase at every other analysis point (10 km spacing), and with the fitted 90°- and 180°-periodic signal displayed at 800 m, 1000 m, 1500 m and 2000 m, for two example lines corresponding to profile A and B in Fig. 1, respectively. Similar figures of the remaining lines can be found in the Supplementary Material, Fig. S 2–S 6.

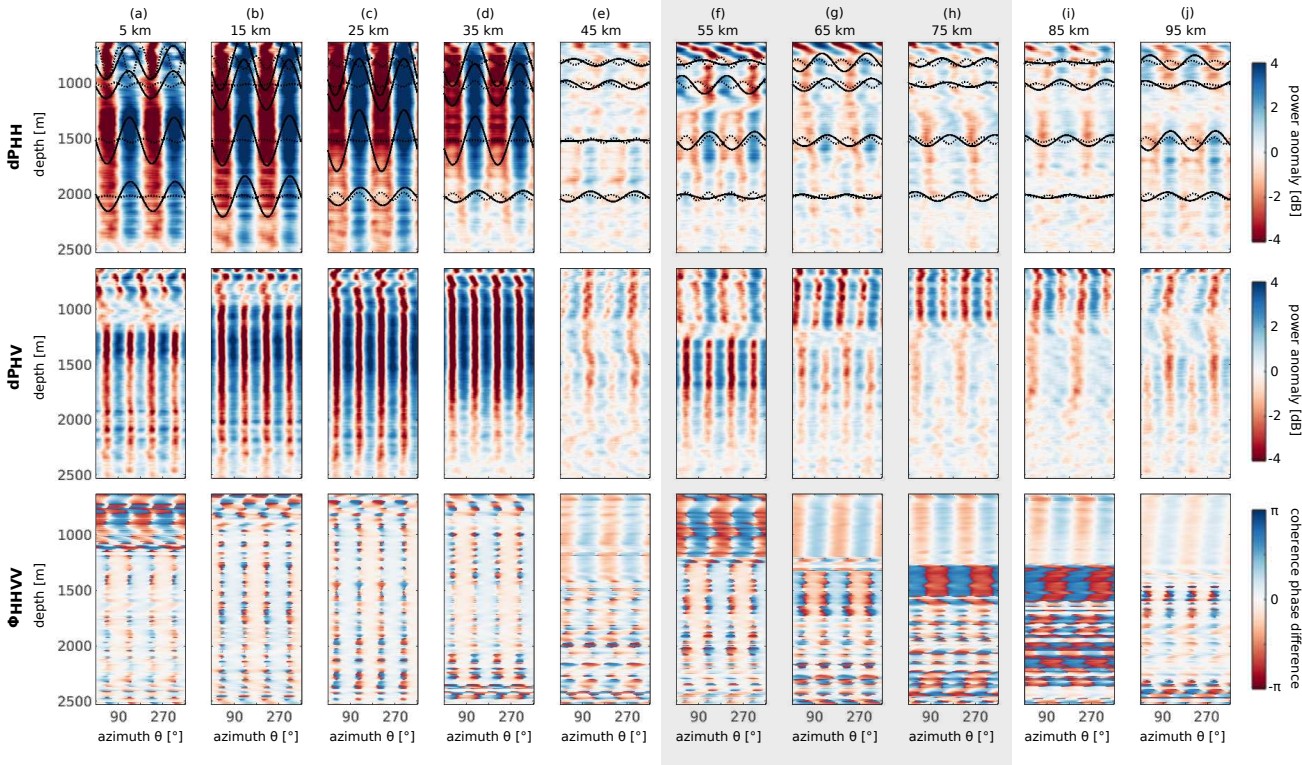

**Figure 6.** Synthesized azimuthal radar response for profile A: top panels show the co-polarized power anomaly dP$_{HH}$ with the fitted 90°- and 180°-periodic sine curve shown as dotted and solid lines at depths of 800 m, 1000 m, 1500 m and 2000 m. Middle panels show the cross-polarized power anomaly dP$_{HV}$, and bottom panels show the coherence phase difference $\phi_{HHVV}$. The $x$-axis on each panel shows the azimuth rotating clockwise from True North. Light gray background (55–75 km) indicates locations along the survey line within 3 km distance from shear margins, and panels with white background (5–45 km and 85–95 km) are located inside the ice stream.

The 90°- and 180°-periodic signal components of the fitted curve are displayed at the corresponding depths in the dP$_{HH}$ panels. A map view of the strength and orientation of the 180°-periodic scattering effect at those same depths is shown in



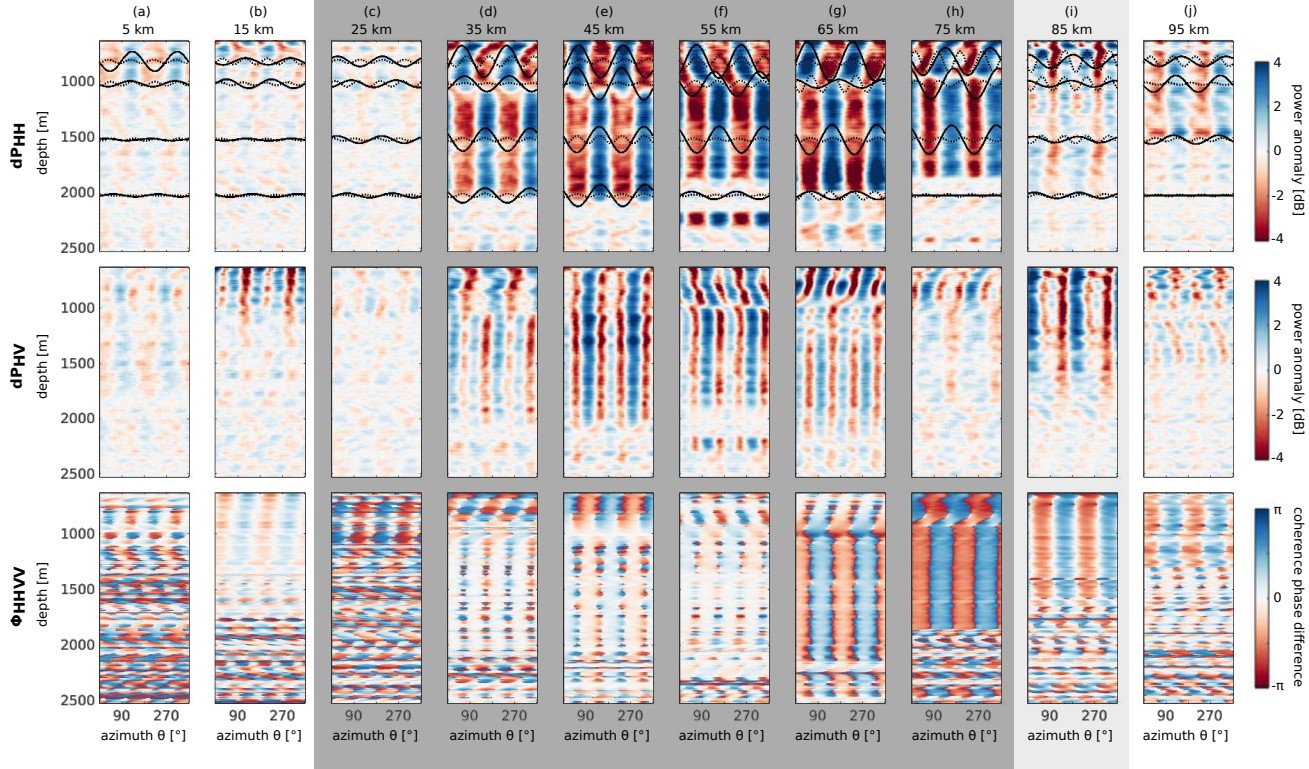

**Figure 7.** Synthesized azimuthal radar response for profile B: top panels show the co-polarized power anomaly dP$_{HH}$ with the fitted 90°- and 180°-periodic sine curve shown as dotted and solid lines at depths of 800 m, 1000 m, 1500 m and 2000 m. Middle panels show the cross-polarized power anomaly dP$_{HV}$, and bottom panels show the coherence phase difference $\phi_{HHVV}$. The $x$-axis on each panel shows the azimuth rotating clockwise from True North. White background (5–15 km and 95 km) indicates locations inside the ice stream, light gray (85 km) is less than 3 km from a shear margin and dark gray (25–75 km) is outside the ice stream.

Fig. 8, together with the apparent horizontal eigenvalue difference ($\Delta\lambda$) derived from HH–VV travel-time differences. The latter represents depth-averages and is based on cross-correlation of the two co-polarized traces parallel and perpendicular to the driving direction. For driving directions not aligned with the COF principal axes this results in apparently smaller eigenvalue difference which doesn't necessarily imply lower anisotropy. For methods detail on deriving the apparent eigenvalue difference from travel times see Gerber et al. (2023) and Zeising et al. (2023).

The periodicity in the CoPE indicates whether anisotropic scattering or birefringence is dominating the radar response. Of all the analyzed points and depths, 80 % are dominated by anisotropic scattering and for 57 % the 180°-periodicity is twice as strong as the 90°-periodicity. The importance of scattering increases with depth. The lowest scattering dominance is found close to the shear margins and at intermediate depths (1000 m, 1500 m), where 59 % of the analyses within less than 3 km from the shear margins are dominated by a 180° periodicity.







**Figure 8.** Orientation and amplitude of anisotropic scattering at a depth of (**a**) 800 m, (**b**) 1000 m, (**c**) 1500 m and (**d**) 2000 m. The orientation (depicted by line orientation) and strength (depicted by size) of scattering is obtained from the amplitude $A_{180}$ and phase $\varphi_{180}$ of the 180°-periodic component in Eq. (20) which has been fitted to $dP_{HH}$. White-to-blue colors in the background indicate depth-averaged apparent horizontal eigenvalue difference derived from HH–VV travel-time differences.



The scattering direction in the ice stream center is oriented between 75° and 85° from the surface flow direction, i.e. 5–15° offset from flow-perpendicular, where scattering is strongest at depths of 1000 m and 1500 m. This misalignment is consistent with the azimuth shift between modeled and observed data in Fig. 4, and suggests that the eigenvectors in the EastGRIP core are not perfectly aligned with the surface flow direction. Near the shear margins, scattering becomes less dominant and

the orientation rotates, which is particularly visible in profile A, following close to the southwestern shear margin in Fig. 8. This rotation of scattering direction goes in hand with an apparent decrease in horizontal anisotropy derived from travel-time differences between along-track and cross-track polarizations. Outside the shear margins, the scattering orientation is close to flow-parallel at shallow depths (Fig. 8a), but rotates by approximately 90° for larger depths (Fig. 8b,c), notably in profile B in in the northwestern corner of Fig. 8 and Fig. 7 at km 35–75.

This reversal of scattering direction is also visible in the HH–VV power difference of the radargrams, demonstrated in Fig. 9, which shows that the power difference clearly follows the ice-sheet stratigraphy. Positive (blue) values indicate increasing return power perpendicular to the driving direction, while negative (red) values indicate increased power parallel to the driving direction. For along-flow profile sections inside the ice stream (e.g. A1–A2, B1–B2, B4–B5) power differences tend to be positive in most parts of the ice column, indicating that the power return approximately perpendicular to ice flow (HH) is up

to 10 dB stronger. Profile parts that were recorded perpendicular to ice flow (e.g. A2–A3, B2–shear-margin, B7–B1) show a negative power difference, confirming that more energy is scattered in the direction perpendicular to flow (in this case VV). A notable change occurs at the 11.4 ka isochrone, marking the Wisconsin-Holocene climatic transition. Inside the ice stream (e.g. the 0–40 km section of profile A and the 0–10 km section of profile B) this transition is marked by an increasing HH–VV power difference, while outside the shear margins (20–70 km of profile B) the sign of the power difference is reversed. In this

same section of profile B, a folded unit between the 65.8 ka and 74.7 ka isochrones is also characterized by different scattering properties and a reversed scattering direction compared to the overlying ice layers. Sections A3–A4, B2–B3 and B5–B6 show alternating signatures of positive and negative power difference, particularly pronounced in Holocene ice, which is a result of birefringence-induced beat signatures (Gerber et al., 2023). In both profiles there is an approximately 200 m thick horizontally isotropic echo-free basal zone.

The 90° periodicity in the cross-polarized power anomaly is constant throughout most of the first 35 km of profile A (Fig. 6), with only small fluctuations above 1200 m depth. A notable azimuth shift occurs at a depth of 1200–1400 m between 55 km and 75 km, i.e. the XPE is shifted by up to 45°. XPE azimuth shifts also appear particularly pronounced in profile B at 55–65 km and depths between 1000 m and 1300 m, indicating a change in COF orientation (Fig. 7).

The coherence phase shows the 90°-periodic dipole nodes which, in contrast to CoPE nodes in $dP_{HH}$, are easy to recognize

as individual nodes regardless of the presence of anisotropic scattering, which only reduces the angular width of these nodes (Ershadi et al., 2022). Anisotropic scattering is, for instance, particularly strong at 5–55 km, and 95 km in profile A (Fig. 6), and 35–55 km of profile B (Fig. 7). The small vertical spacing between the nodes in most parts of the radargrams indicates that strong horizontal anisotropy is present in the survey area. Distinguishing the individual nodes is, however, not straightforward and determining the degree of horizontal anisotropy from the coherence phase is left for future work.





**Figure 9.** Difference in power return of co-polarized modes (HH–VV). Panel (**a**) and (**b**) show profiles A and B in Fig. 1. Turns in the radar lines are marked with $A_1$–$A_4$ and $B_1$–$B_7$ respectively and the corresponding locations are indicated in Fig. 1. The driving direction is from left to right. The gray triangles mark the positions of the shear margins outlined in Fig. 1. Profile B is missing some data at ∼65 km.





## 5 Discussion

We observe the presence of both anisotropic scattering and birefringence effects at EastGRIP and in radar lines recorded within a radius of 50 km from the EastGRIP drill site, extending over the entire ice-stream width and beyond the shear margins. Notably, birefringence is most pronounced near the shear margins and at shallow depths, but is often superimposed by anisotropic scattering which clearly dominates the azimuthal response in the ice-stream interior and exterior, and in particular in ice units dating back to the Wisconsin period. In the following we first discuss the origin of anisotropic scattering, then examine the relationship between observed backscatter properties and climate-induced ice characteristics, and finally explore the implications of our findings for inferring COF type and orientation.

### 5.1 Origin of anisotropic scattering

Three potential mechanisms have previously been identified as being most likely causes for anisotropic scattering (Drews et al., 2012): 1) elongated air bubbles, 2) directional layer roughness, and 3) small-scale variations in COF with depth. We next discuss each process separately.

### 5.1.1 Air bubbles

Air bubbles form near the surface during the transformation of firn into ice. As they get buried under additional snow and ice layers, these bubbles are compressed, resulting in a reduction in their diameter with increasing depth and deformation under a deviatoric stress regime. The amount, size and shape of bubbles can vary between layers deposited under different atmospheric conditions (Svensson et al.), leading to stratification on larger scales which may affect radio waves. Eventually, bubbles disappear as they transform into clathrate hydrates (e.g. Miller, 1969; Ohno et al., 2004), a transition that occurs in the range of 500 to 1000 m depth in the EastGRIP ice core (Stoll et al., 2021), and exhibits only marginal variation across the Greenland ice sheet (Pauer et al., 1999; Kipfstuhl et al., 2001; Neff, 2014). Bubbles tend to be elongated in the direction of dilatational strain (Hudleston, 1977) and can be a proxy of local strain rates (Alley and Fitzpatrick, 1999), which according to findings by Drews et al. (2012) can cause anisotropic scattering with increased return power in the direction of bubble elongation. The flow regime at EastGRIP is dominated by along-flow extension due to acceleration, and flow-transverse compression from lateral inflow (Stoll, 2019; Westhoff et al., 2021; Gerber et al., 2023). Air bubbles are expected to be slightly elongated in the flow direction at EastGRIP, although detailed ice-core studies concerning bubble shapes are still ongoing at the time of writing. Highly elongated bubbles are usually found in rapidly shearing layers with orientation (near-)parallel to the shear zone (Hudleston, 1977; Russell-head and Budd, 1979), i.e. bubbles can be expected to be most elongated in and parallel to the NEGIS shear margins. Inside the ice stream and at depths between 1000 m and 1500 m, our observations show a dominant scattering orientation near-perpendicular to the surface ice-flow direction, which is opposite to the theoretical scattering caused from bubbles elongated along flow. Near the shear margin, a scattering rotation is observed, which, in principal, agrees with a rotation in bubble shape towards parallel to the shear margins. However, the scattering amplitudes are smaller in the proximity of the margins compared to the ice-stream center, which we would not expect if it was caused by bubble elongation. Outside



the shear margins, scattering directions change abruptly with depth at the Wisconsin–Holocene transition, which is hard to explain with bubbles being elongated in orthogonal directions between those ice units.

While it is theoretically possible for bubbles to counteract other mechanisms of anisotropy and cause along-flow scattering in shallow depths outside the ice stream, three key arguments suggest that they play a negligible role in the depths observed by our radar system. Firstly, the spatial distribution of scattering directions and amplitudes are not aligned with expected scattering caused by bubble elongation in the corresponding strain regimes, although this argument is weakened by lack of direct observations. Secondly, the number of bubbles and their size tend to decrease with depth, and smaller bubbles are generally more spherical than larger ones (Alley and Fitzpatrick, 1999). If air bubbles were affecting anisotropic scattering, we would expect the anisotropy to gradually change with depth, which is not confirmed by our observations. Lastly, anisotropic scattering occurs throughout most of the ice column, while at the same time its strength and directionality clearly follows the ice-sheet internal horizon stratigraphy. Although bubble size and shape could in principle be following stratigraphy for example due to influences of impurities, bubbles cannot be the major cause of scattering at depth because they 1) mostly appear in the upper 1000 m of the ice column, and 2) disappear at around the same depth, regardless of ice age. We therefore dismiss elongated air bubbles as major mechanism for anisotropic scattering observed by our radar system.

### 5.1.2 Layer roughness

Evidence for azimuth-dependent internal layer roughness is found in the visual stratigraphy in the EastGRIP core (Westhoff et al., 2021), and directional interface roughness has previously been discussed as a potential mechanism for anisotropic scattering (Drews et al., 2012). The roughness amplitudes visible in the visual record are on the order of centimeters or less, but the observation of potentially present larger-wavelength folds are restricted by the ice-core diameter. Previous studies of roughness in glaciological contexts have focused on the ice surface (e.g. Long and Drinkwater, 2000; Nolin et al., 2002; van der Veen et al., 2009; Segal et al., 2020) and basal roughness (e.g. Hubbard et al., 2000; Eisen et al., 2020; Tang et al., 2022). In principle, interface roughness affects the radar signal through a transition from specular reflection to a more diffuse scattering and wave depolarization when the roughness amplitudes are of the same order of magnitude as the radar wavelength (Peters et al., 2005; Giannopoulos and Diamanti, 2008). For anisotropic roughness, higher backscatter can be expected in the direction perpendicular to the folding axis (Bateson and Woodhouse, 2004; Bartalis et al., 2006). The visual stratigraphy from the Wisconsin ice at EastGRIP shows a higher interface roughness perpendicular to the surface flow direction (Westhoff et al., 2021), i.e. the folding axes are parallel to ice flow due to transversal flow-compression and along-flow extension in the ice-stream center. Assuming that this small-scale roughness also prevails on larger scales relevant for our RES system, as recently shown by Jansen et al. (2024), scattering would lead to increased return power perpendicular to flow, and is consistent with our field observations inside the ice stream. If the directional layer roughness is a result of ice dynamics, in particular lateral strain, layers can be assumed to be more smooth and anisotropic scattering to be less pronounced outside the ice stream. The HH–VV power difference in profile B (Fig. 9) is indeed slightly higher inside the ice stream between 1300 m and 1550 m depth in the first 25 km of the profile compared to the same depth outside. However, this is not consistently so, as for ice younger than 11.4 ka, as well as between 38 ka and 52.7 ka, the power difference is higher outside (at 40 km) than inside the ice stream (at




5 km) of profile B. Additionally, while differences in scattering amplitudes between ice from different climate periods could in principle occur due to different folding amplitudes related to viscosity differences, the reversed directionality of anisotropic scattering between Holocene and Wisconsin ice north of the NW-shear margin would require a remarkably different strain history between those ice units if it were to be explained by ice-flow induced layer roughness.

### 5.1.3 Small-scale vertical variation of COF

The COF eigenvalues in Fig. 3a show a higher variability with depth for $\lambda_y$ (taken to be perpendicular to flow) than $\lambda_x$ (taken to be flow-parallel), leading to a reflection coefficient of up to 15 dB and potential anisotropic scattering with increased return power perpendicular to ice-flow. This is in agreement with our field observations at EastGRIP and the HH–VV power difference of roughly 10 dB in the ice-stream interior. The simulations in Fig. 3 using COF-derived anisotropic scattering show a slightly stronger co-polarized power anomaly at a depth of 1400 m (Holocene–Wisconsin transition) also observed in Fig. 9. While the simulations done here agree quite well with radar-based observations near EastGRIP, it is worth keeping in mind their limitations: A major difficulty in measurements of physical properties in ice cores is that the core orientation is usually lost since the ice core is free to rotate as it is traveling up the borehole. The EastGRIP core has thus been cut in random directions and the true orientation of the COF is unknown. Subsequently, the core has been partially re-oriented between 1375 m and 2120 m depth with a method based on the visual stratigraphy of cloudy bands (Westhoff et al., 2021). For the oriented samples, the smallest COF eigenvector $v_1$ points at roughly 30° East, near-parallel to the surface-flow direction. We assumed that this is the case for the entire COF record, but scattering directions in large parts of the ice-stream center are misaligned with a flow-perpendicular reference by 5-15° (see Fig. 8), which has also been observed in an independent analysis by Nymand et al. (2024). Additionally, azimuthal variations in XPE in the observed and synthesized RES response suggest some sort of COF rotation which is not captured by the model assuming constant eigenvector directions. Additionally, the vertical resolution of the COF record is not high enough to thoroughly understand the small-scale variations and its impact on scattering: while abrupt eigenvalue changes can cause reflections, and hence, anisotropic scattering, this might not be the case for smoother transitions. This might also be the reason why the scattering amplitudes derived from the COF are generally larger than in the observations, since the vertical resolution of the COF dataset is quite coarse.

In spite of these unknowns, differences in the scattering properties between ice units from different climatic periods are most completely explained by changes in the COF and its variability with depth, as both the temperature and impurity content affect the crystal size, shape, and COF development (e.g. Faria et al., 2002; Song et al., 2006; Fan et al., 2020). The XPE rotations in Fig. 4, Fig. 6 and Fig. 7 coincide with the 11.4 ka isochrone depth traced in Fig. 9, and further suggests that a major COF change happens at the Wisconsin-Holocene climatic transition. This observation has also been made in other ice-core analyses (Montagnat et al., 2014) and in polarimetric RES studies. Drews et al. (2012) and Ershadi et al. (2022) found at EPICA Dronning Maud Land (EDML) in Antarctica, that the sign of the reflection ratio changes at the Holocene–Wisconsin transition concurrent with changes in fluctuations of the horizontal eigenvalues with depth, while Martín et al. (2023) observed comparable changes even across multiple climate transitions at the Beyond EPICA drill site BELDC (where ice-core analysis is pending) as well as at South Pole. By ruling out other potential mechanisms for anisotropic scattering and by agreement





between COF-models and RES observations, we suggest that changes in the small-scale COF fluctuations with depth are the most likely cause for the anisotropic scattering observed in the NEGIS onset.

An important distinction must be made between horizontally anisotropic COFs and COFs that lead to anisotropic scattering. Horizontally anisotropic COFs are characterized by a large difference in their horizontal eigenvalues, causing strongly birefringent properties, and are typically found in dynamic areas of ice sheets. Examples of COFs with strong horizontal anisotropy

include vertical girdles often found in flank flow (e.g. Fitzpatrick et al., 2014) or horizontal single maxima which develop in shear zones (e.g. Thomas et al., 2021; Gerber et al., 2023). The scattering property, however, seems to arise from variations in the directional relative permittivities (i.e. $\varepsilon'_x$, $\varepsilon'_y$) and not from the COF type itself, i.e. the noise of the COF distribution rather than its type. In other words, a horizontal single maximum can be highly anisotropic without causing anisotropic scattering if it is perfectly constant with depth, while a weak girdle, in principle, can cause stronger anisotropic scattering when the horizontal

eigenvalues fluctuate independently from each other with depth.

## 5.2 Relation between anisotropic signatures, climate- and strain-induced properties of the ice

The HH–VV power difference in Fig. 9 reveals ice units with distinct scattering properties, including a basal echo-free zone characterized by isotropic scattering, the transition between Holocene and Wisconsin ice (marked by the 11.4 ka isochrone), and folded units with reversed scattering orientation in profile B. Preliminary COF analysis of the deepest part of the EastGRIP

ice core shows a transition from a vertical girdle to a vertical single maximum at around 2475 m depth (Stoll, 2024, pers. communication), approximately 200 m above a muddy base. Transferring this information to the radar observations, it could be possible that the basal zone with isotropic and overall weak return power in profile A and B, which was estimated to be 200 m thick, could mark a similar COF change from the vertical girdle observed into a vertical single maximum. Such a change is in fact often found at the base of glaciers and ice sheets (e.g. Hooke, 1973; Herron and Langway, 1982; Tison et al., 1994;

Cuffey et al., 2000a).

In Sermeq Kujalleq/Jakobshavn Isbrae, Horgan et al. (2008) found a distinct boundary in seismic reflectivity, which they interpreted as the transition between Holocene and Wisconsin ice. Similarly, Wang et al. (2018) found several layers with different seismic scattering properties in Antarctica, which correspond to units from different climatic periods. This would be in accordance with our radar observations, where the transition is marked by an increased scattering amplitude in Wisconsin

ice compared to Holocene ice within the NEGIS interior, and a reversed scattering orientation outside. A similar reversal of scattering orientation is observed in the folded ice between the 65.8 ka and 74.7 ka isochrones in profile B (Fig. 9). This unit is characterized by intact and continuous internal reflection horizons and weaker anisotropic scattering with the same directionality as Holocene ice, while the overlying Wisconsin layers and folding axes exhibit reversed scattering orientation. The age of this folded ice unit remains uncertain, as deep isochrones, particularly the 74.7 ka isochrone, could not be traced

across the shear margin without ambiguities, and because this unit has been affected by folding processes. Ice from cold periods, such as the Wisconsin, consists of smaller ice crystals and has a higher impurity content, both of which lead to softening for deformation compared to ice from warm periods like the Holocene and Eemian (Paterson, 1991; Cuffey et al., 2000b; Faria et al., 2014b, a). These mechanical differences likely influence COF evolution and may result in variations in vertical COF



fluctuations that affect scattering strength and orientation. It is therefore possible that the reversed scattering orientation in the folded ice is due to similar mechanisms observed at the Holocene–Wisconsin transition, suggesting this ice unit could have formed during the Eemian period. In that case, the ice stratigraphy would have been disrupted and inverted during the folding process, similar to observations of folded ice in the GRIP and GISP2 ice cores (Suwa et al., 2006), and in the NEEM ice core where ice from the Eemian period has been folded and part of the stratigraphy has been overturned multiple times (Dahl-Jensen et al., 2013).

The implications of our observations are important for glaciological research and paleoclimate reconstruction. Distinguishing between ice units based on their scattering properties using quad-polarized radars enhances our understanding of ice-sheet dynamics and structure. For example, mapping ice units with different micro-scale properties which affect anisotropic scattering, such as basal shear zones, can improve ice-sheet models by integrating depth-dependent variations in mechanical properties – an aspect typically assumed constant throughout the ice column. Scattering properties following the ice-sheet stratigraphy could also be used in automatic routines for tracing internal horizons. Moreover, if scattering properties can be linked to climate-induced microstructure, quad-polarized radars can aid identifying inverted stratigraphy, which is crucial for paleoclimatic studies and the site-selection of ice-core drilling efforts. It is left for future studies to investigate how anisotropic scattering and climatic transitions relate in other areas of ice sheets, and in how far these scattering properties can be used to reliably map the extent and microstructural properties of ice from different climatic periods.

## 5.3 Implications on inferring COF type and orientation

Cross-polarized power anomalies (dP$_{HV}$) can indicate COF orientation, in particular in combination with the phase coherence gradient (Jordan et al., 2019) but can be challenging when the COF rotates with depth (Ershadi et al., 2022). In our analysis, the rotation of XPE in some parts (Fig. 4, 6, 7) suggest some sort of COF rotation. However, the interpretation is not straightforward due to integrated path effects (Zeising et al., 2023), as a rotation of, say, 45° does not necessarily imply a 45° rotation of the COF axes. The direction of anisotropic scattering can give additional insights into the COF orientation when other scattering origins than COF can be ruled out. The 180° periodic signal in the ice-stream center is rotated clockwise from the flow-perpendicular axes by 5–15°, suggesting that the eigenvectors are not perfectly aligned with the flow as been commonly assumed (Westhoff et al., 2021; Gerber et al., 2023). A recent study by Nymand et al. (2024) shows that the COF orientation can be derived from double reflections in the same RES dataset as used in this study. Their results indicate that the COF in the ice-stream center is rotated around 12° clockwise from the surface flow direction, and is approximately aligned with the scattering axes observed in the central flow line here. A notable rotation of the scattering direction is also shown in the second half of profile A (55–100 km in Fig. 6, Fig. 8), located in the vicinity of the shear margin, where the scattering axes are rotated 20–70° clockwise, with a tendency towards stronger rotation at larger depths. An eigenvector rotation is also found by Nymand et al. (2024) in this same profile. The scattering rotation goes in hand with an apparent decrease of horizontal anisotropy inferred from HH–VV travel-time differences, which can be explained by a COF-rotation away from the polarization directions of the radar, rather than an actual decrease in anisotropy. The agreement between scattering orientation and eigenvector orientation derived by Nymand et al. (2024) gives confidence that scattering can be attributed to COF orientation. Using scattering as a proxy for



COF orientation has the advantage of being independent of depth and strength of anisotropy, and observable at any orientation
of the quad-polarized measurement, while double reflections could only be used in a fraction of the RES data.

While scattering orientation can act as an independent measure for the orientation of COF principal axes when other scatter-
ing sources can be ruled out, it can also complicate efforts to derive COF strength from RES measurements. Previous studies
used airborne RES surveys with standard co-polarized antenna configuration to derive the horizontal eigenvalue difference
from the vertical spacing of CoPE nodes (Young et al., 2021; Gerber et al., 2023). The presence of strong anisotropic scattering
expands CoPE nodes in the vertical (e.g. Ershadi et al., 2022) which makes the identification of individual nodes challeng-
ing while also affecting the azimuth angle at which CoPE appear strongest. Anisotropic scattering could be one reason why
this method proofed to be difficult inside the NEGIS (Gerber et al., 2023). Similar challenges appear in analyses of CoPE in
polarimetric RES measurements. Horizontal eigenvectors and differences in horizontal eigenvalues can also be derived from
the coherence phase which is more robust to anisotropic scattering, but is limited in its ability to detect abrupt COF changes
(Jordan et al., 2019). Ultimately, inverse methods as suggested by Ershadi et al. (2022) are a valuable tool to approximate
the full orientation tensor of the COF. However, well-defined initial conditions are crucial, and the inversion is challenging
when COF rotation with depth is significant. Decomposing the co-polarized power anomalies into its 180°- and 90°-periodic
components can help constraining the initial conditions in both scattering magnitude and the presence of COF rotation with
depth for such inversion efforts in the future.

## 6  Conclusions

We used curve-fitting methods to analyze the relative importance of anisotropic scattering and birefringence on the azimuthal
power response in ground-based quad-polarized RES data collected in the NEGIS onset region. We found that the 180° periodic
effect of anisotropic scattering dominates the co-polarized power anomaly in 80% of the analyzed locations and depths, while
the 90° periodic signal of birefringence tends to be only dominant in depths of less than 1000 m and near the shear margins.
We conclude that small-scale COF fluctuations, i.e. its variance or noise, with depth is the most likely cause of anisotropic
scattering and that the scattering direction is likely aligned with one horizontal eigenvector. Scattering properties strongly
follow the ice-sheet stratigraphy, indicating potential to use quad-polarized measurements to identify ice units with different
scattering properties, e.g. basal zones or ice from cold/warm climatic periods.

Our results are in agreement with previous studies which found anisotropic scattering being dominant over birefringence
in highly dynamic areas, while the opposite is observed in slow-moving locations of ice sheets. This leads to challenges in
deciphering the COF strength and direction from co- and cross-polarized extinction, and we leave a full inversion of quad-
polarimetric data as proposed by Ershadi et al. (2022) for future work. While we are confident that anisotropic scattering is
mostly related to COF here, it remains unclear how other physical parameters like impurity content and crystal size affect these
scattering mechanisms in different ice units. Continuous high-resolution COF measurements of ice cores as well as larger-scale
airborne polarimetric RES surveys would be beneficial in understanding the scattering properties of climatic transitions and
mapping their distribution in ice sheets.



*Code and data availability.* Codes used for data analysis are available on github (https://github.com/tamaragerber/anisotropicScattering). Datasets will be available online upon publication.

## Appendix A: Propagation matrix

The full form of the propagation matrix $\mathbf{E_P}$ is

$$\mathbf{M_P} = \frac{\exp(jk_0 z)^2}{(4\pi z)^2} \times \prod_{i=1}^{N}[\mathbf{R}(\theta_{N+1-i})\mathbf{T}_{N+1-i}\mathbf{R}'(\theta_{N+1-i})] \times \mathbf{R}(\theta_i)\mathbf{S}_i\mathbf{R}'(\theta_i)\prod_{i=1}^{N}[\mathbf{R}(\theta_i)\mathbf{T}_i\mathbf{R}'(\theta_i)] \tag{A1}$$

## Appendix B: Supplementary information

Supplementary figures are available in a separate document.

*Author contributions.* The RES data was recorded by DS, DAL, NFN and TAG, and processed by NFN. TAG designed and carried out the analyses and wrote the manuscript with contributions of all co-authors. All co-authors revised the manuscript.

*Competing interests.* One of the co-authors has been an editor for TC.

*Acknowledgements.* This research has been supported by the Villum Investigator Project IceFlow (grant no. 16572) and by Novo Nordisk Foundation grant NNF23OC0081251. EastGRIP is directed and organized by the Center for Ice and Climate at the Niels Bohr Institute, University of Copenhagen. It is supported by funding agencies and institutions in Denmark (A. P. Møller Foundation, University of Copenhagen), USA (US National Science Foundation, Office of Polar Programs), Germany (Alfred Wegener Institute Helmholtz Center for Polar
and Marine Research), Japan (National Institute of Polar Research and Arctic Challenge for Sustainability), Norway (University of Bergen and Trond Mohn Foundation), Switzerland (Swiss National Science Foundation), France (French Polar Institute Paul-Emile Victor, Institute for Geosciences and Environmental research), Canada (University of Manitoba) and China (Chinese Academy of Sciences and Beijing Normal University). Radar development was supported by funding from the University of Alabama. The authors would like to thank the EGRIP logistic support and field personnel, and in particular Claus Birger Sørensen, Prasad Gogineni, and Drew Taylor for their technical support
in the field.



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
