# Peer review of "Anisotropic Scattering in Radio-Echo Sounding: Insights from Northeast Greenland"

_EGUsphere, 2024_

## Referee Comment (RC1)

**Review of:** Anisotropic scattering in radio-echo sounding: insights from Northeast Greenland
**Submitted to:** The Cryosphere
**Reviewer:** Nicholas Holschuh

**General Comments:**

I want to start my review by saying that the data and analyses in this paper are amazing. This would be one of the first published datasets capable of unambiguously identifying the role for physically (rather than chemically) controlled radar scattering at shallow depths in the ice sheet. The data show clearly that the physical properties of the glacier vary as a function of depth (/age) and location (/glaciological context) in ways that were surprising and informative. These data could change our understanding of microphysical processes playing out within the ice column, and change the way we use radar to understand the structure and dynamics of ice sheets. The fact that different layer packages exhibit qualitative differences in the polarization dependence of scattering means that radar might be capable of uniquely identifying packets of ice across Greenland and Antarctica ***even when they are discontinuous***. This could enable significantly improved estimates of the depth-age structure of ice sheets from radar data, and provide a valuable constraint for ice sheet models. The analysis was robust, and the figures present an incredible amount of data. Overall, I highly recommend this paper for publication.

However, even as someone who has thought deeply about the effects of ice-crystal anisotropy in measured radar backscatter, I found that reading the paper involved a very high cognitive load. At the sentence level, the writing is clear, but at the paragraph and section level, there are many places where the structure gets in the way of the narrative. In the technical and line-item comments below, I highlight places where I stumbled, was confused, or had questions, and provide some recommendations for how the authors might make the text more accessible.

In addition to those structural recommendations, I have one more significant criticism that I think should be addressed. In the main text of the manuscript, the authors fully redescribe the model derived by Fujita et al., (2006) for propagation and scattering, as well as the approach to full-azimuth synthesis from quad-polar data and the calculation of coherence phase from Ershadi et al. (2022). Even though the authors directly invite the reader to skip over that text, I think it goes against best-practices to fully restate the work of other authors and present it in this way. It discourages readers from going back to (and ultimately citing) the original works, and it introduces the possibility that errors might creep into the literature, as we play a game of academic "telephone" that has the potential to corrupt or misrepresent original ideas. As important as the ethical and practical implications of removing that text, I actually find much of the current text in section 3 of this manuscript distracting, as it takes quite a bit of thinking (even as an expert reader) to work through it despite the fact that the forward modeling of the EGRIP data contributes very little to the major conclusions of the paper. I recommend that the restatement of Fujita et al. and Ershadi et al.'s works be mostly removed from the text, and you refer the reader to the original manuscripts for the details.

With that change, I would strongly recommend this paper for publication. Very cool work!

**Technical Comments:**

Here I provide two high-level recommendations that I think might strengthen the work. To improve access to the key conclusions of the paper, I would (a) try to shorten your discussion of the forward model at EGRIP and (b) work to shorten and focus your paragraphs, especially later in the manuscript. I'll provide examples of why those changes might help below.

When I finished reading this paper, my primary takeaways were the following:

- Below 1000 m, anisotropic scattering induces larger variability in measured backscattering (as a function of polarization) than birefringence. Birefringence only plays a larger role in modifying measured back-scatter in the shear margin.
- While several mechanisms have been proposed for anisotropic scattering (including scattering from rough interfaces and elongate bubbles), small-scale variability in ice fabric is the mechanism most consistent with your observations.
- The nature of the anisotropy seems to vary systematically with ice age, and may be related to (and therefore indicative of) the climate conditions at the time of deposition.
- While cross-polarized extinction is related to the COF, path-effects make it difficult to use deep extinction nodes to determine the local fabric orientation. Anisotropic scattering does not suffer from that same limitation, and could be used to uniquely determine the fabric eigenvector directions through the full ice column.

I think these are the things you hoped I would get out of the paper, which is a testament to the current manuscript! But as a reviewer, I am required to be a patient reader, and I came into this paper with significant expertise – not all of your readers will be so patient. None of those conclusions (except maybe the role of fabric in driving anisotropic scattering) really rely on your forward model at EGRIP. And yet, pages 6-12 of the manuscript are devoted to its derivation and description, all of which is text that requires very deep understanding of anisotropy, wave-splitting, and polarization changes to fully process. That is a lot for the reader to have to get through before they arrive at the empirical analysis which is the primary basis for the conclusions of the paper. I think that section 3 could be made much shorter -- once you've motivated the fact that anisotropic scattering and birefringence have different periodicity and that it is possible to synthesize the full azimuthal response using quad-pol data, you could (in theory) present figure 6 and figure 9, which (in my view) are the basis for most of the conclusions of the paper. You really want your reader to get to the content currently on pages 15-18, and I'm not sure every reader will make it through the earlier sections as currently drafted.

Throughout the document, ideas that might be interesting to a broad glaciological audience are intermingled in paragraphs with quite technical radioglaciology. I appreciate how thorough the technical descriptions are; that text is useful for someone like me who might want to build on your analysis. But I think the document could be more approachable if you clearly separate technical details from broader takeaways, and help the reader understand why you've presented the technical details you have. For example, the first paragraph of the introduction describes the goal of the paper but also lists four survey design strategies. These survey design details are probably of low interest to a non-radioglaciology audience, and coming so early in the paper, they are not clearly situated in the broader narrative. Why

should the reader care about these different survey types, especially at this point in the work? If you were to move lines 23-32 to their own paragraph later in the introduction (for example, following the paragraph ending at line 67), you motivate why the reader should care much more clearly. Then your introduction would be: (P1) Polarimetric radar tells us about the ice sheet structure, (P2) the signal is affected by birefringence (P3) and anisotropic scattering, (P4) there are several ways to collect the data, and (P5) this is what past studies have shown.

This is just one example of how slight structural changes might reduce the intellectual burden on the reader. In the line-item comments, I provide examples of other places where I think you can simplify your structure like this to make the document a bit more accessible. But independent of my specific recommendations, I would encourage you to read through the document with the goal of shortening and focusing paragraphs, pulling the more important takeaways out of deep discussions of technical detail.

**Line-Item Corrections:**

| | |
|---|---|
| Page #: 1
Line #: 3-8 | Something to consider -- I think these three sentences (starting "Although both…") can be made more succinct and combined into one, which gets the reader to your results more quickly. |
| Page #: 2
Line #: 23-32 | As I mentioned in the technical comments, I think it makes more sense to pull these sentences out of your first paragraph (which is meant to provide the highest level introduction and motivation) and move them to after line 67. That first paragraph could naturally end with the sentences spanning 32-34, and I think it will improve the flow of the introduction. |
| Page #: 3
Line #: 56 | I think "affecting the overall return power" is a slightly confusing addition to this sentence. It is clearer if it reads "Volume scattering is caused by small-scale inhomogeneities in the physical properties of the glacier, such as air bubbles, dust particles, or impurities." |
| Page #: 3
Line #: 58 | The phrase "also observed in the optical range" assumes that the reader knows that you've been thinking about scattering in the microwave range up to this point, which isn't explicitly stated (at least in this paragraph). Might be useful to rephrase to something like "Anisotopric volume scattering is significant in the optical range..." or something like that, and then make clear in the next sentence that you are talking about RES data: "Surface scattering of radio waves occurs..." |
| Page #: 3
Line #: 69-72 | These two sentences present another example of content that feels unrelated to the larger paragraph, and belongs somewhere else in the paper. If I were to summarize the purpose of this paragraph, it is to introduce the limitations of previous radar approaches to estimating ice fabric. But the sentences stating "The COF type" and "In-situ measurements" don't contribute to this idea at all. Without realizing it, these seeingly minor narrative culs-de-sac really increase the difficulty in reading and understanding a work, so minimizing them really helps me as a reader. |

| | |
|---|---|
| Page #: 4
Line #: 98-124 | Section 2 is great. Succinct and clear! |
| Page #: 6
Line #: 132 | For the reasons I described in the general comments, I think lines 132-184 and 199-207 should be removed in favor of a clear set of references to the original equations in Fujita et al., 2000, 2006, and Ershadi et al., 2022. I think you want to use this space to emphasize to the reader why, exactly, you did this analysis, and be as brief as possible. Something like "Here we show that synthesis from quadpol data faithfully reproduces the results of the turning circle, and both datasets capture the primary features we would expect based on forward-modeling the radio-wave propagation problem using direct fabric measurements collected from the EGRIP core." Much of the rest of the description of the forward model results (lines 215-237) doesn't feel clearly situated in the narrative. The reader doesn't know yet why they should care about any of the things you observe. I would cut them or move them to later in the document when they become relevant for interpreting the sources of scattering. |
| Page #: 9
Line #: 181 | While I think this section should be cut entirely, if you keep this text, this sentence is missing a concluding clause. |
| Page #: 14
Line #: 261-264 | Why do you estimate the eigenvalue difference from travel-time differences rather than integrated phase difference from your synthesized coherence phase, using the orientation where you know you are aligned with the COF? That would let you account for lack of alignment of your driving direction and the COF. It would at least be interesting to compare those results. |
| Page #: 15
Line #: Fig 8 | You don't present any of the radargrams crossing the eastern shear margin, but there seems to be a lot of variability in the eigenvalue differences presented (I assume, associated with the folding there?). It would be worth interpreting for the reader those signals, because they appear really coherent across the flow-perpendicular lines. Is it that the system has limited capacity to capture the folding because of its narrow beam pattern, and so those are artifacts? I didn't see any mention of focusing in your processing steps either -- how might that affect interpratation of those signals? I can't tell if I should think of them as interesting signal or artifactual in nature, a result of the limitations of the system, and it would be nice to clarify that somewhere for the reader. |
| Page #: 15
Line #: Fig 8 | I'm also interested in that really intense eigenvalue difference found between A-2 and A-3 at the southern end of your survey. Is that real? That looks like (in Figure 9) a region where there is very low signal. |
| Page #: 16
Line #: 271-272 | How should I think about "flow parallel" vs. "flow perpendicular" when thinking about anisotropic scattering? Is it fair to just thinkg about it as "the principle fabric axes are shifted 15 degrees clockwise(?) from the modern flow field"? I spent a long |

time trying to figure out if I should care that it is perpendicular rather than parallel, and I realized I wasn't exactly sure what that distinction meant.

| | |
|---|---|
| Page #: 16
Line #: 293-294 | I find this phrasing a bit confusing -- without echoes, how do you know it is isotropic? A bit more detail on your thinking here would be helpful. |
| Page #: 17
Line #: Fig 9 | I think it would be helpful for the reader if you highlight and describe those features due to birefringence seen from 50-100 km in the shallow part of Profile A and from 20-40 in the shallow part of Profile B. I think those signals are really interesting and represent what most folks had been looking for in co-polarized data up to this point, so it is useful to highlight and describe them here. |
| Page #: 18
Line #:
Section 5.1 | This section justifies one of the major conclusions of the paper, but sections 5.1.1/2/3 would benefit from a slight reorganization and additional paragraph breaks. Right now, these sections have paragraphs with 10-15 sentences which tend to wander back and forth between ideas. The cleaner the structure you can provide for the reader, the better. I was going to suggest some specific paragraph breaks, but I think sentences need to get moved around in a way that makes the ideas more clearly separable before you subdivide. |
| Page #: 19
Line #: 360-361 | Can you substantiate this statement with a bit more detail? I found this statement counterintuitive, and was having trouble understanding why the opposite wouldn't be true. |
| Page #: 20
Line #: 389-390 | These two sentences both start with "additionally". I would restructure this paragraph in general (noting my comment on section 5.1), but try to avoid that kind of repitition. |
| Page #: 21
Line #: 407-415 | This paragraph is really important context for the reader to have, and because it is not a discovery of this paper, I think it should come much earlier in the paper. It might be most appropriate when you introduce the forward model and the 90/180 degree periodicity. Highlighting that the two methods for azimuthal variability have distinct implications for fabric before the reader spends a long time obeserving the differences in fabric would be very helpful. |
| Page #: 21
Line #: 416-417 | Again, I find this phrasing odd -- "echo free zone characterized by … scattering". What exactly do you mean here? |

| Page #: 22
Line #: 439-444 | I'm not sure the conclusion you're drawing here (that because the Eemian ice at NEEM was overturned the signal in the polarimetric data here indicates local ice is overturned) is correct. My mental model of how you get changes in sign of the HH-VV power difference is the following: |
|---|---|

[Figure]

When you have greater variability in the permittivity associated with one polarization than another, you have greater scattering, and therefore a higher power. But imagine you took a section of B and inverted it. You wouldn't then produce a change in the sign of the HH-VV difference; that would actually require a 90 degree rotation of ice in B. Is that consistent with your mental model, or am I missing something? I would say either explain in more detail how inversion would manifest in anisotropic scattering or remove this section here (and the reference to it in the abstract).

| Page #: 23
Line #: 481 | "proved" rather than "proofed" |
|---|---|

| Page #: 23
Line #: 489 | At present, you don't fully explain why this is the case for the shear margin – do you think it is because the fabric variability is orientation independent and therefore anisotropy in scattering is weak, or because the effects of birefringence are particularly strong? A bit more discussion of this somehwere in text would be helpful. |
|---|---|

As a reminder – I really enjoyed this paper. I think it is a really valuable contribution and want to see it have as much impact as it deserves!

---- Nick

---

## Referee Comment (RC2)

**Review of "Anisotropic Scattering in Radio-Echo Sounding: Insights from Northeast Greenland"**

This manuscript presents new quad-polarized ground-based radio-echo sounding data from the Northeast Greenland Ice Stream. The authors synthesize the full polarimetric azimuthal radar response from the quad-pol data and then analyze how the azimuthal periodicity of the power difference between polarizations varies spatially and with depth. Their analysis demonstrates that anisotropic scattering is present at almost every depth below 630 m and that birefringence only dominates the polarimetric response near the surface. They also show that there is an abrupt shift in the orientation of the anisotropic scattering at the Holocene-Wisconsin ice boundary, suggesting that radar polarimetry may be a powerful tool for identifying ice from different climatic periods.

Overall, I think this is an excellent piece of work! The demonstration of the ground-based quad-pol radar data alone is very exciting, since this opens the door for applying polarimetric analyses of fabric over larger areas, compared to current approaches with ApRES. The implication that we may be able to identify ice units from different climatic periods based on their polarimetric response could be incredibly useful for placing broad age constraints on discontinuous ice units or in places where we do not have easy connections to dated ice cores. I think this paper will also be a key in forcing the community to acknowledge and account for the impacts of anisotropic scattering on estimates of COF anisotropy from birefringent effects. I have no major concerns with the technical aspects of the manuscript. Most of my comments are directed towards ways that the authors can streamline and clarify what is currently a very dense technical paper that can sometimes obscure its exciting core results with tangents in the narrative.

**Major Comments:**

**[1]** Since most of the conclusions are based around the idea that anisotropic scattering signatures will have 180-degree periodicity vs 90-degree periodicity for birefringence, it would be incredibly helpful to the reader to spend a few sentences describing why this is the case. I think I've convinced myself that it must be due to the integrated two-way path propagation effects for birefringence, but discussing these ideas explicitly would be great for readers who are not deep experts in radar measurements of fabric.

**[2]** Section 3.1 rederives a polarimetric scattering model from Fujita et al. (2006) to model the radar response to COF at EGRIP. In the end, this model does not seem to be central to the paper's major conclusions, especially since there are some significant differences between the modeled and measured polarimetric responses. To my reading, the model provides a very broad sanity check on the measured polarimetric response and is briefly used to justify the argument that COF variability drives the deep anisotropic scattering. Considering this, I think that the discussion of the model can be much more concise and probably just point readers to Fujita et al. (2006) model, which will lighten the mental load for readers.

**[3]** Almost a quarter of the paper (pages 6-12) is devoted to convincing the reader that the quad-pol synthesis can be trusted. I actually do not think that level of detail is necessary and bogs the reader down in a long technical discussion before they ever get to the main methods of the paper. The quad-pol synthesis method is firmly rooted in the governing equations of electromagnetics, has been demonstrated multiple times with quad-pol ApRES in our field, and is routinely used outside the field

in other radar applications. If anything, the turning circle may be less reliable because it aggregates the polarimetric response over a series of radar footprints that do not fully overlap and may be subject to effects from layer slope, for example. Therefore, I think it is totally sufficient to just cite the quad-pol synthesis method and make this section as concise as possible. To me at least, the main value of the comparison with the turning circle is to demonstrate that the quad-pol instrument has sufficient radiometric calibration and phase synchronization across channels, a motivation which was surprisingly not mentioned in the paper.

**[4]** You might consider breaking out the discussion of the sinusoidal fit into its own section. This is the main analysis method that is used throughout the rest of the paper, so it would be very valuable to give it a clear emphasis rather than burying it at the end of the discussion on the quad-pol synthesis.

**[5]** Overall, I would encourage you to think carefully about the specific purpose(s) of presenting the turning circle-synthesis-model comparison and be explicit about this purpose at the beginning of the section. Then limit the technical details and discussion of the comparison to the most salient points that are needed to support that purpose.

**[6]** I found Figures S7-S13 really helpful for following the discussion of how anisotropic scattering vs. birefringence varied with depth. If at least one of those plots could be added to the main paper, I think that would be very valuable. For example, perhaps adding a fourth panel to Figure 6 (or Figure 7) with the amplitude of each sinusoid as a function of depth for each location a-j.

**In-Line Comments:**

Line 40 – since dual or quad-pol satellite SAR is also used in many glaciological applications and has a different viewing geometry, it would be good to specify something about radio-echo sounding here.

Line 58-60 – the mention of optical anisotropic scattering seems unnecessary since this entire paper is about radio frequency measurements.

Figure 2 – you might consider marking ice flow directions and adding labels for inside vs. outside the ice stream and the shear margins in this image, just to help the reader who otherwise has to flip back and forth with Figure 1 quite a bit.

Line 110-111 – I would recommend adding a few comments on the final horizontal resolution and trace spacing after processing, and perhaps why SAR focusing was not employed.

Figure 3 – the colors in panel a are hard to distinguish due to the black outlines, particularly the purple.

Line 196-197 – where does this reflection ratio come from and what justifies this choice?

In Figures 6-7, it would be fantastic to add markers in some way for the same isochrones which are shown in Figure 9. This would help the reader better visualize how changes in the azimuthal response with depth are related to stratigraphic units and age. It would also be very helpful to have some annotations showing the key features that a reader should take away from the $dP_{HV}$ and $\phi_{HHVV}$ plots.

They only get 3-4 sentences in the discussion, and I found it a bit hard to track the key points that I should take away from these plots.

Figure 8 – I find the high frequency spatial variations in the apparent horizontal eigenvalue difference hear the eastern shear margin very notable. Do you have an idea of what might cause this? Is this "real" or an artifact of low signal to noise ratio and the vertical "streaking" that we commonly see in shear margin radargrams due to dipping layers and/or damage?

Lines 293-294 – how can we know that there is isotropic scattering if the region is "echo-free"? I would guess this just reflects the isotropy of thermal (e.g. white Gaussian) noise rather than something about the ice sheet?

Section 5.1.2 – I'm not entirely convinced by this discussion on the direction of folding vs. scattering. In the citations in this section (Bartalis et al., 2006 for example), anisotropic scattering occurs because the radar is side-looking and so in one orientation the folds act like corner reflectors (high backscatter) and in the other orientation they do not (low backscatter). It's less clear to me how this would work for a nadir-looking radar sounder. My first thought is that you might have stronger co-polarized scattering parallel to the folding axis in the same way that backscatter from a half cylinder can (in some cases) be strongest when the wave polarization is aligned with the long axis of the cylinder, rather than perpendicular to it (see for example (Scanlan et al., 2022)). Anyway, this would further support your argument that roughness is likely not the cause of the anisotropic scattering you observe, but it is worth thinking through the mechanisms in this discussion in the context of radar sounders a bit more.

399 – is there any evidence for a COF-induced reflection at this transition (e.g. an englacial layer in the radiostratigraphy marking what appears to be a quite abrupt transition)?

Lines 407-415 – this is a very interesting and important piece of the discussion! I will admit I found it a bit hard to visualize how the COF would be changing with depth to achieve the anisotropic scattering, and I wonder if you might consider adding a conceptual diagram. Maybe some idealized Schmidt diagrams as a function of depth to explain how you envision the fabric changing?

Line 441 – I am not entirely following how the folding/overturning of stratigraphy would lead to this expression of anisotropic scattering – perhaps you can expound on this a bit? (Maybe this is something else that could be part of an idealized fabric as a function of depth sketch?)

---

## Author Comment (AC1)

**Author's response to review by Nicholas Holschuh**

**January 12, 2025**

Dear Dr. Holschuh,

Thank you for your detailed and constructive feedback on our manuscript. We appreciate your positive remarks on the quality and significance of our data and analyses, as well as your recognition of the broader implications of our findings for understanding radar scattering and ice sheet microphysical processes. It is gratifying to know that you value our work.

Your suggestions for improving the manuscript's accessibility and readability have been especially helpful. We acknowledge your critique of specific sections, including the discussion of the previously published model. We included these details with the intention of making the text more accessible and understandable for readers unfamiliar with the prior work. However, we also understand that this approach has unintended consequences, as it may lead to misinterpretations and potential errors in the literature in addition to disrupting the narrative of the paper. Thank you for raising awareness of this important aspect of scientific communication. We agree that clarity and proper attribution are essential, and we have carefully addressed your comments in the detailed, point-by-point response below.

Once again, we thank you for your valuable insights.

On behalf of all authors,

Tamara

**1 Technical comments**

**1. Shortening the Discussion of the Forward Model (Pages 6-12)**

In the revised manuscript we have removed the detailed model description and instead refer the readers directly to Fujita et al. (2006). In addition to that, we have also removed the detailed discussion on the forward model, and the comparison between forward model, turning circle and synthesized response from Section 3 to the Supplementary Material, which contains all information to reproduce Fig. 3 and Fig. 4. With these changes, Section 3 is now considerably shorter and focuses on the method of determining the orientation and strength of scattering vs. birefringence effects.

**2. Paragraph Structure and Focus**

We appreciate your suggestion to simplify and streamline paragraphs to make the manuscript more accessible, particularly for a broader glaciological audience. We have revised the manuscript and tried

to shorten and simplify sections which contain unnecessary details or which are repetitive, and we restructured others for a more logical line of thought. We hope that this way the revised manuscript is more accessible to a wider glaciological audience.

**2 Line-Item Corrections:**

- **Page 1, Line 3-8:** Something to consider – I think these three sentences (starting "Although both. . . ") can be made more succinct and combined into one, which gets the reader to your results more quickly.

  Thanks. We have revised the text as follows **(line 3–5 in revised manuscript)**:

  *We use curve-fitting techniques to evaluate the relative contributions of anisotropic scattering and birefringence in quad-polarized ground-based RES measurements from the Northeast Greenland Ice Stream (NEGIS), identifying their dominance and orientation across depths of 630–2500 m.*

- Page 2, Line 23-32: As I mentioned in the technical comments, I think it makes more sense to pull these sentences out of your first paragraph (which is meant to provide the highest level introduction and motivation) and move them to after line 67. That first paragraph could naturally end with the sentences spanning 32-34, and I think it will improve the flow of the introduction.

  We have removed these details here and moved the overview of previous polarimetric observations further down the section, line 62–73 in revised manuscript.

- **Page 3, Line 56:** I think "affecting the overall return power" is a slightly confusing addition to this sentence. It is clearer if it reads "Volume scattering is caused by small-scale inhomogeneities in the physical properties of the glacier, such as air bubbles, dust particles, or impurities."

  We have removed "affecting the overall return power" and streamlined the section for clarity. The revised text now reads (line 40–45):

  *Anisotropic scattering describes the directional dependence of ice's scattering properties, which causes variations in signal intensity based on the orientation of the antenna. Scattering of radio waves in ice sheets arises from two main mechanisms: volume scattering, driven by small-scale inhomogeneities within the ice, and surface scattering, caused by reflections at internal interfaces (e.g., Langley et al., 2009; Drews et al., 2012). Volume scattering includes contributions from air bubbles, dust, impurities, and anisotropic features such as elongated air bubbles or small-scale fluctuations in horizontal permittivities related to COF (Drews et al., 2012).*

- **Page 3, Line 58:** The phrase "also observed in the optical range" assumes that the reader knows that you've been thinking about scattering in the microwave range up to this point, which isn't explicitly stated (at least in this paragraph). Might be useful to rephrase to something like "Anisotopric volume scattering is significant in the optical range..." or something like that, and then make clear in the next sentence that you are talking about RES data: "Surface scattering of radio waves occurs..."

  We have removed the reference to the optical range, as it is not relevant to this paper. Additionally, we clarified at the start of the paragraph that we are focusing exclusively on radio waves (line 40–42):

  *Scattering of **radio waves in ice sheets** arises from two main mechanisms: volume scattering, driven*

*by small-scale inhomogeneities within the ice, and surface scattering, caused by reflections at internal interfaces (e.g., Langley et al., 2009; Drews et al., 2012).*

- **Page 3, Line 69-72:** These two sentences present another example of content that feels unrelated to the larger paragraph, and belongs somewhere else in the paper. If I were to summarize the purpose of this paragraph, it is to introduce the limitations of previous radar approaches to estimating ice fabric. But the sentences stating "The COF type" and "In-situ measurements" don't contribute to this idea at all. Without realizing it, these seeingly minor narrative culs-de-sac really increase the difficulty in reading and understanding a work, so minimizing them really helps me as a reader.

  We have removed these two sentences from this paragraph to improve focus and readability.

- **Page 4, Line 98-124:** Thanks!

- **Page 6, Line 132:** For the reasons I described in the general comments, I think lines 132-184 and 199- 207 should be removed in favor of a clear set of references to the original equations in Fujita et al., 2000, 2006, and Ershadi et al., 2022. I think you want to use this space to emphasize to the reader why, exactly, you did this analysis, and be as brief as possible. Something like "Here we show that synthesis from quadpol data faithfully reproduces the results of the turning circle, and both datasets capture the primary features we would expect based on forward-modeling the radio-wave propagation problem using direct fabric measurements collected from the EGRIP core." Much of the rest of the description of the forward model results (lines 215-237) doesn't feel clearly situated in the narrative. The reader doesn't know yet why they should care about any of the things you observe. I would cut them or move them to later in the document when they become relevant for interpreting the sources of scattering.

  We have significantly shortened Section 3 to improve focus and narrative flow. Specifically, we have:

  1. Removed the detailed description of the used model.

  2. Relocated Fig. 3 and Fig. 4 to the Supplementary Information, along with the necessary details to reproduce the model results.

  3. Condensed the text to focus on the key explanation of the curve-fitting method, illustrated with the modeled, turning circle, and synthesized response.

  These changes streamline the section, focusing on why the analysis was conducted, while less immediately relevant details are provided later in the manuscript or in the Supplementary Information.

- **Page 9, Line 181:** While I think this section should be cut entirely, if you keep this text, this sentence is missing a concluding clause.

  This section has been removed.

- **Page 14, Line 261-264:** Why do you estimate the eigenvalue difference from travel-time differences rather than integrated phase difference from your synthesized coherence phase, using the orientation where you know you are aligned with the COF? That would let you account for lack of alignment of your driving direction and the COF. It would at least be interesting to compare those results.

We chose to estimate the eigenvalue difference using travel-time differences instead of the integrated phase difference because of the challenges presented by strong anisotropy in our study area. The coherence method, which works well in areas with low anisotropy (as shown in Jordan et al., 2022; Young et al., 2021), becomes problematic in ice streams with strong anisotropy where scatterer depth differences between polarization directions exceed the radar's range resolution, leading to loss of coherence. For our ultra-wideband radar system, the coherence method can in theory only be applied above 350 m depth, where the scatterer-depth difference is within range resolution (Zeising et al., 2024). As recently suggested by Zeising et al. (2024), reducing bandwidth through zero-padding could perhaps extend the depth at which the coherence method can be applied for the data presented here, but requires data re-processing and would considerably impact signal strength.

Given these challenges, we decided that using travel-time differences was the most effective approach for providing context to the COF orientation derived from anisotropic scattering, while also showing the limitations of that method. We have clarified this reasoning in the revised manuscript (line 187–192):

*The integrated phase difference can in theory be used to derive horizontal eigenvalue differences, a method which works reliably in low-anisotropy areas (e.g. Jordan et al., 2022; Young et al., 2021). However, for strong COF anisotropy the depth differences between reflections of opposite polarization directions exceeds the radar's range resolution, leading to loss of coherence (Zeising et al., 2024). Although the phase coherence method could not be applied successfully to our dataset for deriving COF eigenvalues, reprocessing the data to reduce radar bandwidth might improve its applicability in future efforts, albeit at the cost of signal strength (Zeising et al., 2024).*

- **Page 15, Fig 8:** You don't present any of the radargrams crossing the eastern shear margin, but there seems to be a lot of variability in the eigenvalue differences presented (I assume, associated with the folding there?). It would be worth interpreting for the reader those signals, because they appear really coherent across the flow-perpendicular lines. Is it that the system has limited capacity to capture the folding because of its narrow beam pattern, and so those are artifacts? I didn't see any mention of focusing in your processing steps either – how might that affect interpratation of those signals? I can't tell if I should think of them as interesting signal or artifactual in nature, a result of the limitations of the system, and it would be nice to clarify that somewhere for the reader.

Thank you for this detailed observation. Indeed, the observed changes in anisotropy appear to align with folds mapped by Jansen et al. (2024) (now added to the figure background). To clarify the nature of these signals, we have re-evaluated the calculation of the eigenvalue difference ($\Delta\lambda$) to ensure that it reflects real features rather than processing artifacts.

In the original analysis, eigenvalue differences were derived by automating the process to cross-correlate traces in a sliding window, identifying the lag with maximum alignment. $\Delta\lambda$ was then calculated only for correlations exceeding 0.6. However, in regions with steeply inclined layers, reflections are sparse, and high correlations can occasionally result from noise rather than true signals. Similarly, if there are too few reflections or if the reflections are too shallow, the eigenvalue difference becomes difficult to estimate accurately, leading to potential over- or underestimation of anisotropy.

To address these limitations, we have refined the criteria for deriving $\Delta\lambda$:

1. Correlations must exceed 0.6, and reflections must have amplitudes above the noise floor, as

      derived from areas near the shear margins where no reflections are visible.

2. Eigenvalue differences are now calculated only when there are at least 10 reflections, with at least one extending deeper than 1200 m.

These stricter criteria improve the reliability of the results and reduce the influence of noise or insufficient reflections. The eigenvalue differences in near the shear margin are not considered very reliable with this new criteria and their variability an artifact of the strong folding in that area. We have clarified these points in line 194–201 of the revised manuscript:

*The eigenvalue difference was determined using an automated process that measures the travel-time difference between the HH and VV traces. Specifically, the cross-correlation of each trace pair was calculated within a 20 m sliding window to estimate the time delay between signals. Linear regression was then applied to correlated reflections to obtain the depth-averaged apparent eigenvalue difference (for method details see Gerber et al., 2023). The uncertainty of this method increases when only shallow reflections are available or when the number of reflections is low. To ensure reliability, we included only results where at least ten internal reflections could be correlated with a correlation coefficient above 0.6, and where at least one reflection lies below 1200 m depth. Results were discarded where these criteria were not met, particularly in areas with steeply dipping internal layers near shear margins.*

For completeness, we also added radargrams of the remaining lines as HH-VV power difference to the Supplementary Information.

- **Page 15, Fig 8:** I'm also interested in that really intense eigenvalue difference found between A-2 and A-3 at the southern end of your survey. Is that real? That looks like (in Figure 9) a region where there is very low signal.

Eigenvalue differences near the shear margin are indeed expected to increase due to the dynamic nature of the ice in this region, as shown by models and shallow ice cores (Gerber et al., 2023). However, as you correctly noted, the eigenvalue difference observed between A-2 and A-3 at the southern end of the survey is based on only a few relatively shallow reflections. This makes it less reliable compared to areas where more and deeper reflections are available.

As mentioned in our response to your previous comment, we have tightened the criteria for confidently deriving eigenvalue differences. Under these stricter criteria, this region does not meet the necessary confidence threshold. Consequently, this part has been removed in the updated figure to ensure the presented data reflect only robust results.

- **Page 16, Line 271-272:** How should I think about "flow parallel" vs. "flow perpendicular" when thinking about anisotropic scattering? Is it fair to just thinkg about it as "the principle fabric axes are shifted 15 degrees clockwise(?) from the modern flow field"? I spent a long time trying to figure out if I should care that it is perpendicular rather than parallel, and I realized I wasn't exactly sure what that distinction meant.

Your interpretation is correct: the principal fabric axes are shifted approximately 5–15° clockwise from the modern surface flow field, which has previously been assumed to be aligned based on reconstructed orientation of the EastGRIP ice core (Westhoff et al., 2021).

The terms "flow-parallel" vs "flow-perpendicular" are not essential for understanding this concept since eigenvectors are per definition perpendicular to each other. We have removed "flow-perpendicular" to avoid potential confusion and here simply modified the text as follows (line 202–203):

*The scattering direction in the ice stream center is oriented between 95° and 105° clockwise from the surface flow direction at depths of 1000 m–1500 m where scattering is strongest.*

Please also note that we corrected 75–85° to 95-105° clockwise rotation (75-85° would be anticlockwise).

To clearly distinguish between observations and interpretation, we moved the description of the implications of this observation to Section 5.3 (line 369–380):

*The direction of anisotropic scattering can give independent indication of the COF orientation when other scattering origins than COF can be ruled out. The 180° periodic signal in the ice-stream center is rotated clockwise from the flow direction by 95–105°, suggesting that the eigenvectors are not perfectly aligned with the surface flow as has been commonly assumed (Westhoff et al., 2021; Gerber et al., 2023), a conclusion which was also reached by Nymand et al. (2024) using double reflections to derive COF orientation from the same dataset as this study. A notable rotation of the scattering direction is also shown in the second half of profile A (55–100 km in Fig. 4, Fig. 6), located in the vicinity of the shear margin, where the scattering axes are rotated 20–70° clockwise, with a tendency towards stronger rotation at larger depths. Here again, Nymand et al. (2024) found similar results. The COF rotation here explains the apparent decrease in horizontal anisotropy derived from travel-time differences, which does not represent the true anisotropy in case of misalignment of COF axis and radar wave polarization. The agreement between scattering COF orientation derived by Nymand et al. (2024) further supports our conclusion that scattering can be attributed to COF orientation, which has the advantage of being independent of depth and strength of anisotropy, and observable at any orientation of the quad-polarized measurement.*

- **Page 16, Line 293-294:** I find this phrasing a bit confusing – without echoes, how do you know it is isotropic? A bit more detail on your thinking here would be helpful.

  Thank you for pointing this out. In this context, we were using the term "isotropic" to indicate the absence of anisotropy. However, we understand that this phrasing can be confusing. To avoid misunderstanding, we have removed the term "isotropic scattering" here and similar contexts elsewhere.

- **Page 17, Fig 9:** I think it would be helpful for the reader if you highlight and describe those features due to birefringence seen from 50-100 km in the shallow part of Profile A and from 20-40 in the shallow part of Profile B. I think those signals are really interesting and represent what most folks had been looking for in co-polarized data up to this point, so it is useful to highlight and describe them here.

  We added further description of these beat signatures in line 221–227 of the revised manuscript:

  *Sections A3–A4, B2–B3 and B5–B6 show alternating signatures of positive and negative power difference, particularly pronounced in Holocene ice, which is a result of birefringence-induced beat signatures: Birefringence causes a rotation of the polarization ellipsoid in and out of the profile plane, so the power alternates between being higher parallel (VV) and perpendicular (HH) to the profile direction. These birefringence-induced beat signatures are indicative of the misalignment of radar antennas and COF*

*principal axes. The fact that strong beat signatures are mostly visible at shallow depths might be due to the loss of coherence at larger depths.*

- **Page 18, Section 5.1:** This section justifies one of the major conclusions of the paper, but sections 5.1.1/2/3 would benefit from a slight reorganization and additional paragraph breaks. Right now, these sections have paragraphs with 10-15 sentences which tend to wander back and forth between ideas. The cleaner the structure you can provide for the reader, the better. I was going to suggest some specific paragraph breaks, but I think sentences need to get moved around in a way that makes the ideas more clearly separable before you subdivide.

We have tried to reorganize this section for more clarity. The line of thoughts now is as follows:

**air bubbles:** 1) background of bubble formation and distribution in the Greenland ice sheet, 2) how bubbles elongate and how they are expected to be oriented in and around NEGIS, 3) how this might affect radar return power, which 4) is not consistent with our observations and that's why we reject this as the main cause for anisotropic scattering.

**layer roughness:** 1) observation and expected orientation of layer roughness in the NEGIS, 2) how it potentially affects scattering, 3) why it is not consistent with our observations.

**COF fluctuations:** Having ruled out the above mechanisms we explain in this section how we imagine COF fluctuations causing the observed scattering with an additional illustration (a suggestion by reviewer 2) including two example scenarios: 1) inside the NEGIS, where COF is known from the EastGRIP ice core and 2) in the folded units north of NEGIS.

- **Page 19, Line 360-361:** Can you substantiate this statement with a bit more detail? I found this statement counterintuitive, and was having trouble understanding why the opposite wouldn't be true.

Thanks for being critical here. This statement was supported by the referenced studies. However, as reviewer 2 pointed out, those are based on radars with considerably larger beamwidth, and other studies have suggested the opposite for subglacial cylindrical channels. So it actually is not exactly clear how anisotropic roughness would affect the data from our system. We have clarified this in the text (lines 275–294 of the revised manuscript) but ultimately rule out interface roughness as the primary cause of anisotropic scattering for the reasons provided:

*The effect of directional interface roughness on radar return power is complex. Interface roughness can transition radar signals from specular reflection to more diffuse scattering and wave depolarization when roughness amplitudes are comparable to the radar wavelength (Peters et al., 2005; Giannopoulos and Diamanti, 2008). Studies with side-looking radars have shown that higher backscatter occurs perpendicular to the folding axis, as folds act as corner reflectors (Bateson and Woodhouse, 2004; Bartalis et al., 2006). However, for a nadir-looking radar system with a much narrower beamwidth, this anisotropic scattering mechanism may not operate in the same way. Instead, stronger co-polarized scattering might occur parallel to the folding axis, depending on fold size and radar characteristics (Scanlan et al., 2022).*

*Despite the unclear relationship between folds and anisotropic scattering, we can rule out directional interface roughness as the major source of anisotropic scattering for the following reasons: First, if directional interface roughness results from ice dynamics, particularly lateral strain, we would expect*

*layers outside the ice stream to be smoother, with less pronounced anisotropic scattering. Indeed, the scattering amplitude is generally slightly higher inside the ice stream than outside (Fig.??). However, this pattern is not consistent. For example, scattering amplitudes outside NEGIS in profile B exceed the amplitudes in the ice-stream interior particularly downstream of EastGRIP and in profiles which are not in the ice-stream center (Fig.??a–c). Although roughness outside the current ice stream might be remnants of previous ice-dynamic configurations, the spacial distribution of scattering amplitudes is difficult to be explained by roughness alone, particularly the lower amplitudes towards ice-stream margins where folding amplitudes are known to increase (Jansen et al., 2024). Second, while scattering differences between ice from different climate periods could stem from variations in folding amplitudes associated with viscosity differences, the reversed directionality of anisotropic scattering between Holocene and Wisconsin ice north of the NW shear margin would imply an exceptionally distinct strain history between these ice units if attributed to ice-flow-induced interface roughness, which is unrealistic.*

- **Page 20, Line 389-390:** These two sentences both start with "additionally". I would restructure this paragraph in general (noting my comment on section 5.1), but try to avoid that kind of repitition.

  *Thanks, we restructured this part, and checked for repetitions similar to this.*

- **Page 21, Line 407-415:** This paragraph is really important context for the reader to have, and because it is not a discovery of this paper, I think it should come much earlier in the paper. It might be most appropriate when you introduce the forward model and the 90/180 degree periodicity. Highlighting that the two methods for azimuthal variability have distinct implications for fabric before the reader spends a long time obeserving the differences in fabric would be very helpful.

  *Thanks for this suggestion. We moved it to the introduction to provide the context as early as possible, line 52–61 of the revised manuscript.*

- **Page 21, Line 416-417:** Again, I find this phrasing odd – "echo free zone characterized by ... scattering". What exactly do you mean here?

  *Agreed. We have changed it to simply 'echo-free zone'.*

- **Page 22, Line 439-444:** I'm not sure the conclusion you're drawing here (that because the Eemian ice at NEEM was overturned the signal in the polarimetric data here indicates local ice is overturned) is correct. My mental model of how you get changes in sign of the HH- VV power difference is the following:

[Figure]

When you have greater variability in the permittivity associated with one polarization than another, you have greater scattering, and therefore a higher power. But imagine you took a section of B and inverted it. You wouldn't then produce a change in the sign of the HH-VV difference; that would actually require a 90 degree rotation of ice in B. Is that consistent with your mental model, or am I missing something? I would say either explain in more detail how inversion would manifest in anisotropic scattering or remove this section here (and the reference to it in the abstract).

Thank you for this comment. To clarify, we do not suggest that the anisotropic scattering is reversed *because* of overturned ice, but rather that the observed change in the scattering properties is indicative of different ice units, which could be ice formed under conditions similar to the Holocene (e.g., Eemian ice). If that is the case, it follows that the stratigraphy must be disrupted (and perhaps overturned) because Eemian ice is significantly older than the traced isochrones. As you correctly point out, an inversion of ice strata in section B would not lead to a change in the sign of the HH-VV difference and this is not what we wanted to indicate.

We have now clarified this point in the manuscript by more clearly distinguishing the change in scattering properties as an indicator of a different ice unit, rather than attributing it directly to the effect of folded ice (line 344–351 in revised manuscript):

*In Fig. 8, we proposed a potential mechanism for the reversed scattering pattern, though we do not claim to fully explain the formation of these COF differences. Ice from colder periods, like the Wisconsin, tends to have higher impurity content and smaller crystals, promoting easier deformation compared to ice from warmer periods like the Holocene and Eemian (Paterson, 1991; Cuffey et al., 2000; Faria et al., 2014a,b). The folding of ice itself does not inherently produce a 90° rotation of COF needed to invert the scattering signature. However, changes in the regional ice dynamics could have altered the local strain regime to which the COF adjusts accordingly. The rate and manner of this adjustment may differ between ice units, with Wisconsin ice, having generally higher impurity content and smaller grains,*

*potentially adjusting more rapidly or distinctly than Holocene ice, which could explain the observed scattering differences.*

- **Page 23, Line 481:** "proved" rather than "proofed"

  done.

- **Page 23, Line 489:** At present, you don't fully explain why this is the case for the shear margin – do you think it is because the fabric variability is orientation independent and therefore anisotropy in scattering is weak, or because the effects of birefringence are particularly strong? A bit more discussion of this somehwere in text would be helpful

  Our current interpretation is that the shear margins exhibit limited anisotropic scattering due to the steeply inclined layers, which result in reduced radar return power overall. Although birefringence is a contributing factor, we find that anisotropic scattering is still more significant than birefringence in the shear margin, except at shallow depths.

  We have added a short paragraph in line 381–385 of the revised manuscript:

  *It is worth noting that care should be taken in areas of strongly folded internal stratigraphy, in our study area particularly in the vicinity of the shear margins. Here, the overall return power is decreased because of steep internal layers. Hence the smaller scattering amplitudes do not necessarily imply smaller COF fluctuations with depth, but may simply reflect the overall decreased return power. Although birefringence effects are mostly found to be dominant near the shear margins and at relatively shallow depths, anisotropic scattering still dominates in most of the analyzed cases.*

**References**

Bartalis, Z., Scipal, K., and Wagner, W. (2006). Azimuthal anisotropy of scatterometer measurements over land. *IEEE Transactions on Geoscience and Remote Sensing*, 44(8):2083–2092.

Bateson, L. and Woodhouse, I. (2004). Observations of scatterometer asymmetry over sand seas and derivation of wind ripple orientation. *International Journal of Remote Sensing*, 25(10):1805–1816.

Cuffey, K. M., Conway, H., Gades, A., Hallet, B., Raymond, C. F., and Whitlow, S. (2000). Deformation properties of subfreezing glacier ice: Role of crystal size, chemical impurities, and rock particles inferred from in situ measurements. *Journal of Geophysical Research: Solid Earth*, 105(B12):27895–27915.

Drews, R., Eisen, O., Steinhage, D., Weikusat, I., Kipfstuhl, S., and Wilhelms, F. (2012). Potential mechanisms for anisotropy in ice-penetrating radar data. *Journal of Glaciology*, 58(209):613–624.

Faria, S. H., Weikusat, I., and Azuma, N. (2014a). The microstructure of polar ice. part i: Highlights from ice core research. *Journal of Structural Geology*, 61:2–20.

Faria, S. H., Weikusat, I., and Azuma, N. (2014b). The microstructure of polar ice. part ii: State of the art. *Journal of Structural Geology*, 61:21–49. Microdynamics of Ice.

Fujita, S., Maeno, H., and Matsuoka, K. (2006). Radio-wave depolarization and scattering within ice sheets: a matrix-based model to link radar and ice-core measurements and its application. *Journal of Glaciology*, 52(178):407–424.

Gerber, T. A., Lilien, D. A., Rathmann, N. M., Franke, S., Young, T. J., Valero-Delgado, F., Ershadi, M. R., Drews, R., Zeising, O., Humbert, A., et al. (2023). Crystal orientation fabric anisotropy causes directional hardening of the northeast greenland ice stream. *Nature Communications*, 14(1):2653.

Giannopoulos, A. and Diamanti, N. (2008). Numerical modelling of ground-penetrating radar response from rough subsurface interfaces. *Near Surface Geophysics*, 6(6):357–369.

Jansen, D., Franke, S., Bauer, C. C., Binder, T., Dahl-Jensen, D., Eichler, J., Eisen, O., Hu, Y., Kerch, J., Llorens, M.-G., et al. (2024). Shear margins in upper half of northeast greenland ice stream were established two millennia ago. *Nature communications*, 15(1):1193.

Jordan, T. M., Martín, C., Brisbourne, A. M., Schroeder, D. M., and Smith, A. M. (2022). Radar characterization of icecrystal orientation fabric and anisotropicviscosity within an antarctic ice stream. *Journal of Geophysical Research: Earth Surface*, 127:e2022JF006673.

Langley, K., Lacroix, P., Hamran, S.-E., and Brandt, O. (2009). Sources of backscatter at 5.3 ghz from a superimposed ice and firn area revealed by multi-frequency gpr and cores. *Journal of Glaciology*, 55(190):373–383.

Nymand, N. F., Lilien, D. A., Gerber, T. A., Hvidberg, C. S., Steinhage, D., Gogineni, P., Taylor, D., and Dahl-Jensen, D. (2024). Double reflections in novel polarized radar data reveal ice fabric in the north east greenland ice stream. *ESS Open Archive*.

Paterson, W. (1991). Why ice-age ice is sometimes "soft". *Cold Regions Science and Technology*, 20(1):75–98.

Peters, M. E., Blankenship, D. D., and Morse, D. L. (2005). Analysis techniques for coherent airborne radar sounding: Application to west antarctic ice streams. *Journal of Geophysical Research: Solid Earth*, 110(B6).

Scanlan, K. M., Buhl, D. P., and Blankenship, D. D. (2022). Polarimetric Airborne Radar Sounding as an Approach to Characterizing Subglacial Röthlisberger Channels. *IEEE Journal of Selected Topics in Applied Earth Observations and Remote Sensing*, 15:4455–4467.

Westhoff, J., Stoll, N., Franke, S., Weikusat, I., Bons, P., Kerch, J., Jansen, D., Kipfstuhl, S., and Dahl-Jensen, D. (2021). A stratigraphy-based method for reconstructing ice core orientation. *Annals of Glaciology*, 62(85-86):191–202.

Young, T. J., Schroeder, D. M., Jordan, T. M., Christoffersen, P., Tulaczyk, S. M., Culberg, R., and Bienert, N. L. (2021). Inferring ice fabric from birefringence loss in airborne radargrams: Application to the eastern shear margin of thwaites glacier, west antarctica. *Journal of Geophysical Research: Earth Surface*, 126(5):e2020JF006023.

Zeising, O., Arenas-Pingarrón, Á., Brisbourne, A. M., and Martín, C. (2024). Brief communication: Reduced bandwidth improves the depth limit of the radar coherence method for detecting ice crystal fabric asymmetry. *EGUsphere*, 2024:1–11.

---

## Author Comment (AC2)

**Author's response to review by anonymous Reviewer # 2**

January 12, 2025

Dear Reviewer,

Thank you very much for your thoughtful and constructive feedback on our manuscript! We are grateful to hear that you find our work valuable.

We value your detailed suggestions in making this work more accessible, in particular your input on improving figures and text clarity. Thank you also for being critical on whether specific sections are really necessary to deliver the most important messages. We have carefully implemented your comments in the revised manuscript which is now shortened, more focused and hopefully more accessible to a wider glaciological audience. Please find the point-by-point response below.

Thanks again for taking the time to review this paper.

On behalf of all authors,

Tamara

**1    Major Comments:**

[1] Since most of the conclusions are based around the idea that anisotropic scattering signatures will have 180-degree periodicity vs 90-degree periodicity for birefringence, it would be incredibly helpful to the reader to spend a few sentences describing why this is the case. I think I've convinced myself that it must be due to the integrated two-way path propagation effects for birefringence, but discussing these ideas explicitly would be great for readers who are not deep experts in radar measurements of fabric.

Thanks for this comment. We have added an explication for this at the beginning of Section 3 which hopefully gives the readers some intuition, (line 120–134 in revised manuscript). However, as this characteristics has been known for around five decades now in the radio glaciology community (see Hargreaves, 1978) and discussed in a number of papers, some of which also referenced in this manuscript, we would prefer to refrain from a further detailed description.

*The effects of anisotropic scattering and birefringence can be distinguished by their periodicity of co-polarized power anomalies, $dP_{HH}$ (for definition see Section 1 in the supplementary Information or Ershadi et al., 2022). These patterns differ because the two mechanisms are governed by distinct symmetries.*

*Amplitude variations due to anisotropic scattering originate from variations in scattering properties within the ice that have a two-fold symmetry. This means that the returned signal is strongest in two opposite directions, resulting in a 180° periodicity when co-polarized antennas are rotated.*

*In contrast, birefringence splits the transmitted radar wave into two orthogonal components that travel at different speeds through anisotropic ice. The relative amplitude of these two wave components depends on the orientation of the antennas relative to the COF axes, while the phase difference depends solely on the degree of anisotropy, but is independent on the antenna orientation. Interference between the two wave components modulates the return power. A 90° rotation of co-polarized antennas flips the amplitudes of the two components, but the interference pattern remains unchanged. Therefore, birefringence creates a 90° periodicity in the co-polarized signal.*

*Both mechanisms exhibit 90° periodicity in cross-polarized anomalies, $dP_{HV}$, because the transformation of the polarization state, whether through scattering or birefringent phase shifts, inherently alternates every quarter-wave (90°), driven by the geometry of the polarization ellipse.*

[2] Section 3.1 rederives a polarimetric scattering model from Fujita et al. (2006) to model the radar response to COF at EGRIP. In the end, this model does not seem to be central to the paper's major conclusions, especially since there are some significant differences between the modeled and measured polarimetric responses. To my reading, the model provides a very broad sanity check on the measured polarimetric response and is briefly used to justify the argument that COF variability drives the deep anisotropic scattering. Considering this, I think that the discussion of the model can be much more concise and probably just point readers to Fujita et al. (2006) model, which will lighten the mental load for readers.

We have considerably shortened Section 3 and removed the model equations and refer to Fujita et al., 2006 instead. We moved Fig. 3 and Fig. 4 into the Supplementary Information with the necessary details to reproduce our results in order to improve the readability of the manuscript and focus more on the essential parts. Section 3 now focuses primarily on how we synthesized the polarimetric response and on the curve-fitting method to determine the relative importance of each contribution.

[3] Almost a quarter of the paper (pages 6-12) is devoted to convincing the reader that the quad-pol synthesis can be trusted. I actually do not think that level of detail is necessary and bogs the reader down in a long technical discussion before they ever get to the main methods of the paper. The quad-pol synthesis method is firmly rooted in the governing equations of electromagnetics, has been demonstrated multiple times with quad-pol ApRES in our field, and is routinely used outside the field in other radar applications. If anything, the turning circle may be less reliable because it aggregates the polarimetric response over a series of radar footprints that do not fully overlap and may be subject to effects from layer slope, for example. Therefore, I think it is totally sufficient to just cite the quad-pol synthesis method and make this section as concise as possible. To me at least, the main value of the comparison with the turning circle is to demonstrate that the quad-pol instrument has sufficient radiometric calibration and phase synchronization across channels, a motivation which was surprisingly not mentioned in the paper.

We agree that the comparison with the turning circle is not crucial for the main conclusions of the paper. As mentioned above, we have considerably shortened Section 3 and moved Fig. 4 to the

Supplementary Information. There we also added a sentence regarding the radiometric calibration and phase synchronization across channels.

[4] You might consider breaking out the discussion of the sinusoidal fit into its own section. This is the main analysis method that is used throughout the rest of the paper, so it would be very valuable to give it a clear emphasis rather than burying it at the end of the discussion on the quad-pol synthesis.

This is now the main aspect of Section 3, which has become much shorter.

[5] Overall, I would encourage you to think carefully about the specific purpose(s) of presenting the turning circle-synthesis-model comparison and be explicit about this purpose at the beginning of the section. Then limit the technical details and discussion of the comparison to the most salient points that are needed to support that purpose.

See previous comments.

[6] I found Figures S7-S13 really helpful for following the discussion of how anisotropic scattering vs. birefringence varied with depth. If at least one of those plots could be added to the main paper, I think that would be very valuable. For example, perhaps adding a fourth panel to Figure 6 (or Figure 7) with the amplitude of each sinusoid as a function of depth for each location a-j.

Thank you for this suggestion. We have added another row of panels in both Fig. 6 and Fig. 7 showing the amplitude-depth profiles.

**2  In-Line Comments:**

- **Line 40:** since dual or quad-pol satellite SAR is also used in many glaciological applications and has a different viewing geometry, it would be good to specify something about radio-echo sounding here.

  We specified this as follows (line 27-28 in revised manuscript):

  *For nadir wave propagation, the standard setup in ice-sheet RES, these polarizations lie in the horizontal plane, assuming that one eigenvector is vertical.*

- **Line 58-60:** the mention of optical anisotropic scattering seems unnecessary since this entire paper is about radio frequency measurements.

  We have removed the reference to the optical spectrum.

- **Figure 2:** you might consider marking ice flow directions and adding labels for inside vs. outside the ice stream and the shear margins in this image, just to help the reader who otherwise has to flip back and forth with Figure 1 quite a bit.

  Thanks for this suggestion — we have added flow directions and shear margin locations to Fig. 2.

- **Line 110-111:** I would recommend adding a few comments on the final horizontal resolution and trace spacing after processing, and perhaps why SAR focusing was not employed.

  Thank you - we added the following clarification (line 102–105 in revised manuscript):

*While synthetic aperture radar (SAR) focusing of the RES data could improve some radargram sections, challenges related to irregular tracks and high bandwidth complicated motion compensation, limiting overall improvement. Consequently, SAR focusing was not employed. The final trace spacing after processing is approximately 25–30 m on average, while it is approximately 0.7 m in the turning circle.*

- **Figure 3:** the colors in panel a are hard to distinguish due to the black outlines, particularly the purple.

  Thanks for flagging this. The line thickness has been decreased to improve color visibility. This Figure has also been moved to the Supplementary Information (Fig. S1).

- **Line 196-197:** where does this reflection ratio come from and what justifies this choice?

  We reduced the scattering amplitude derived from COF variations in order to better match the scattering amplitudes observed with the radar. The reflection ratio derived from COF may be too high because of the low sampling rate of COF measurements, so the eigenvalue variations might be 'smoother' and thus the reflection ratio lower than captured by the COF record.

  This section was moved to the supplementary Information, but we've added a brief explanation there (Supplementary Information Section S2):

  *The amplitude of the COF-derived reflection ratio in the model is nearly twice as high as that observed for anisotropic scattering in RES data. This discrepancy may stem from the low sampling rate of COF measurements, which might fail to capture eigenvalue variations that are smoother in reality. Consequently, the actual reflection ratio may be lower than suggested by the COF data. To account for this, we use a value of 0.5 r[dB] for comparison with the RES data.*

- **Figures 6-7:** it would be fantastic to add markers in some way for the same isochrones which are shown in Figure 9. This would help the reader better visualize how changes in the azimuthal response with depth are related to stratigraphic units and age. It would also be very helpful to have some annotations showing the key features that a reader should take away from the $dP_{HV}$ and $\Phi_{HHVV}$ plots. They only get 3-4 sentences in the discussion, and I found it a bit hard to track the key points that I should take away from these plots.

  Thanks for this suggestion, we have implemented the depth of the isochrones in Fig. 9 in Fig. 6–7. However, we find these two Figures now getting quite busy, and therefore hesitate adding further annotations on $dP_{HV}$ and $\Phi_{HHVV}$ panels. Instead we have elaborated on these two aspects a bit more in the text (line 176–192 in the revised manuscript) with reference to the corresponding panels in Fig. 6–7.

- **Figure 8:** I find the high frequency spatial variations in the apparent horizontal eigenvalue difference hear the eastern shear margin very notable. Do you have an idea of what might cause this? Is this "real" or an artifact of low signal to noise ratio and the vertical "streaking" that we commonly see in shear margin radargrams due to dipping layers and/or damage? Thank you for pointing this out. Indeed, these fluctuations are artifacts due to steeply inclined layers, where the correlation between traces is decreased and anisotropy likely underestimated. We've revised the automated algorithm we used to derive eigenvalue difference and implemented stricter thresholds for where the trace correlation

is good enough that we accept the derived eigenvalue difference as quite reliable. We added this in line 194–201 in the revised manuscript:

*The eigenvalue difference was determined using an automated process that measures the travel-time difference between the HH and VV traces. Specifically, the cross-correlation of each trace pair was calculated within a 20 m sliding window to estimate the time delay between signals. Linear regression was then applied to correlated reflections to obtain the depth-averaged apparent eigenvalue difference (for method details see Gerber et al., 2023). The uncertainty of this method increases when only shallow reflections are available or when the number of reflections is low. To ensure reliability, we included only results where at least ten internal reflections could be correlated with a correlation coefficient above 0.6, and where at least one reflection lies below 1200 m depth. Results were discarded where these criteria were not met, particularly in areas with steeply dipping internal layers near shear margins.*

- **Lines 293-294:** how can we know that there is isotropic scattering if the region is "echo-free"? I would guess this just reflects the isotropy of thermal (e.g. white Gaussian) noise rather than something about the ice sheet?

  Thank you for pointing out this confusing notation, we were referring to horizontally isotropic by absence of anisotropy, but removed this term, as 'echo-free basal zone' is sufficiently accurate.

- **Section 5.1.2** I'm not entirely convinced by this discussion on the direction of folding vs. scattering. In the citations in this section (Bartalis et al., 2006 for example), anisotropic scattering occurs because the radar is side-looking and so in one orientation the folds act like corner reflectors (high backscatter) and in the other orientation they do not (low backscatter). It's less clear to me how this would work for a nadir-looking radar sounder. My first thought is that you might have stronger co- polarized scattering parallel to the folding axis in the same way that backscatter from a half cylinder can (in some cases) be strongest when the wave polarization is aligned with the long axis of the cylinder, rather than perpendicular to it (see for example (Scanlan et al., 2022)). Anyway, this would further support your argument that roughness is likely not the cause of the anisotropic scattering you observe, but it is worth thinking through the mechanisms in this discussion in the context of radar sounders a bit more.

  Thank you for being critical on this section and for pointing out the paper by Scanlan et al., 2022. The relation between anisotropic scattering and layer roughness is, admittedly, more complex than what is outlined in the paragraph, and depends on a variety of factors such as radar beamwidth, roughness amplitude and wavelength. I think it is indeed difficult to understand how exactly it would affect our radar return. We have adressed this issue by 1) clarifying that the findings by Bateson and Woodhouse, 2004 and Bartalis et al 2006 are for side-looking radars and 2) that for nadir surveys the scattering direction might be opposite (as found by Scanlan et al, 2022), which 3) makes it unclear how exactly anisotropic roughnesss would affect radar returns.

  However, we still think it is safe to discard interface roughness as the major driver of anisotropic scattering for two reasons: First, the difficulties to explain reversed scattering directions between ice units and second, the only marginal differences between scattering amplitude inside (where roughness is expected to be higher) and outside NEGIS (where roughness is expected to be lower).

  The revised text now is (line 275–294):

*The effect of directional interface roughness on radar return power is complex. Interface roughness can transition radar signals from specular reflection to more diffuse scattering and wave depolarization when roughness amplitudes are comparable to the radar wavelength (Peters et al., 2005; Giannopoulos and Diamanti, 2008). Studies with side-looking radars have shown that higher backscatter occurs perpendicular to the folding axis, as folds act as corner reflectors (Bateson and Woodhouse, 2004; Bartalis et al., 2006). However, for a nadir-looking radar system with a much narrower beamwidth, this anisotropic scattering mechanism may not operate in the same way. Instead, stronger co-polarized scattering might occur parallel to the folding axis, depending on fold size and radar characteristics (Scanlan et al., 2022). Despite the unclear relationship between folds and anisotropic scattering, we can rule out directional interface roughness as the major source of anisotropic scattering for the following reasons: First, if directional interface roughness results from ice dynamics, particularly lateral strain, we would expect layers outside the ice stream to be smoother, with less pronounced anisotropic scattering. Indeed, the scattering amplitude is generally slightly higher inside the ice stream than outside (Fig. 6). However, this pattern is not consistent. For example, scattering amplitudes outside NEGIS in profile B exceed the amplitudes in the ice-stream interior particularly downstream of EastGRIP and in profiles which are not in the ice-stream center (Fig. 6a–c). Although roughness outside the current ice stream might be remnants of previous ice-dynamic configurations, the spacial distribution of scattering amplitudes is difficult to be explained by roughness alone, particularly the lower amplitudes towards ice-stream margins where folding amplitudes are known to increase (Jansen et al., 2024). Second, while scattering differences between ice from different climate periods could stem from variations in folding amplitudes associated with viscosity differences, the reversed directionality of anisotropic scattering between Holocene and Wisconsin ice north of the NW shear margin would imply an exceptionally distinct strain history between these ice units if attributed to ice-flow-induced interface roughness, which is unrealistic.*

- **Line 399:** is there any evidence for a COF-induced reflection at this transition (e.g. an englacial layer in the radiostratigraphy marking what appears to be a quite abrupt transition)?

  We did not identify evidence for a reflector explicitly caused by a change in COF type. Reflections at the Wisconsin-Holocene transition are commonly observed in radar echograms due to differences in ice acidity, which affect conductivity. However, changing impurity content can also cause COF changes which then would appear in radar and seismic data (Horgan et al., 2008). While COF-induced reflections are typically distinguishable by their anisotropic nature, in this case, the strong anisotropic scattering observed in the data may obscure any reflection resulting solely from a COF change.

- **Lines 407-415:** this is a very interesting and important piece of the discussion! I will admit I found it a bit hard to visualize how the COF would be changing with depth to achieve the anisotropic scattering, and I wonder if you might consider adding a conceptual diagram. Maybe some idealized Schmidt diagrams as a function of depth to explain how you envision the fabric changing?

  Based on a suggestion of reviewer 1, we have moved this paragraph to the introduction, in order to provide this context of our analysis early on in the manuscript. Additionally, we have tried to visualize our idea on how anisotropic scattering might be caused by different COF types in two different locations, inside the ice stream, where COF is known at the EastGRIP drill site, and in the folded units north of NEGIS, where the COF is unknown. We hope this additional Figure (Fig. 8 in revised manuscript)

helps readers to understand the concept, even though some of it remains quite speculative.

- **Line 441:** I am not entirely following how the folding/overturning of stratigraphy would lead to this expression of anisotropic scattering – perhaps you can expound on this a bit? (Maybe this is something else that could be part of an idealized fabric as a function of depth sketch?)

We suggest that the difference in anisotropic scattering is related to ice units formed under different (climatic) conditions, but we don't claim that the folding/overturning itself lead to a change in scattering properties. The fact that different units within these large folds outside of NEGIS exhibit distinct scattering properties could indicate that ice from the Eemian period is present between the 65.8 ka and 74.7 ka isochrones within these folds.This would mean that the stratigraphy there is disrupted, and possibly overturned. In that sense, anisotropic scattering can be indicative of overturned stratigraphy, but not so much the result of it per se.

We have addressed this comment in line 344–351 in the revised manuscript:

*In Fig. 8, we proposed a potential mechanism for the reversed scattering pattern, though we do not claim to fully explain the formation of these COF differences. Ice from colder periods, like the Wisconsin, tends to have higher impurity content and smaller crystals, promoting easier deformation compared to ice from warmer periods like the Holocene and Eemian (Paterson, 1991; Cuffey et al., 2000; Faria et al., 2014a,b). The folding of ice itself does not inherently produce a 90° rotation of COF needed to invert the scattering signature. However, changes in the regional ice dynamics could have altered the local strain regime to which the COF adjusts accordingly. The rate and manner of this adjustment may differ between ice units, with Wisconsin ice, having generally higher impurity content and smaller grains, potentially adjusting more rapidly or distinctly than Holocene ice, which could explain the observed scattering differences.*

**References**

Bartalis, Z., Scipal, K., and Wagner, W. (2006). Azimuthal anisotropy of scatterometer measurements over land. *IEEE Transactions on Geoscience and Remote Sensing*, 44(8):2083–2092.

Bateson, L. and Woodhouse, I. (2004). Observations of scatterometer asymmetry over sand seas and derivation of wind ripple orientation. *International Journal of Remote Sensing*, 25(10):1805–1816.

Cuffey, K. M., Conway, H., Gades, A., Hallet, B., Raymond, C. F., and Whitlow, S. (2000). Deformation properties of subfreezing glacier ice: Role of crystal size, chemical impurities, and rock particles inferred from in situ measurements. *Journal of Geophysical Research: Solid Earth*, 105(B12):27895–27915.

Ershadi, M. R., Drews, R., Martín, C., Eisen, O., Ritz, C., Corr, H., Christmann, J., Zeising, O., Humbert, A., and Mulvaney, R. (2022). Polarimetric radar reveals the spatial distribution of ice fabric at domes and divides in east antarctica. *The Cryosphere*, 16(5):1719–1739.

Faria, S. H., Weikusat, I., and Azuma, N. (2014a). The microstructure of polar ice. part i: Highlights from ice core research. *Journal of Structural Geology*, 61:2–20.

Faria, S. H., Weikusat, I., and Azuma, N. (2014b). The microstructure of polar ice. part ii: State of the art. *Journal of Structural Geology*, 61:21–49. Microdynamics of Ice.

Gerber, T. A., Lilien, D. A., Rathmann, N. M., Franke, S., Young, T. J., Valero-Delgado, F., Ershadi, M. R., Drews, R., Zeising, O., Humbert, A., et al. (2023). Crystal orientation fabric anisotropy causes directional hardening of the northeast greenland ice stream. *Nature Communications*, 14(1):2653.

Giannopoulos, A. and Diamanti, N. (2008). Numerical modelling of ground-penetrating radar response from rough subsurface interfaces. *Near Surface Geophysics*, 6(6):357–369.

Hargreaves, N. (1978). The radio-frequency birefringence of polar ice. *Journal of Glaciology*, 21(85):301–313.

Horgan, H. J., Anandakrishnan, S., Alley, R. B., Peters, L. E., Tsoflias, G. P., Voigt, D. E., and Winberry, J. P. (2008). Complex fabric development revealed by englacial seismic reflectivity: Jakobshavn isbræ, greenland. *Geophysical Research Letters*, 35(10).

Jansen, D., Franke, S., Bauer, C. C., Binder, T., Dahl-Jensen, D., Eichler, J., Eisen, O., Hu, Y., Kerch, J., Llorens, M.-G., et al. (2024). Shear margins in upper half of northeast greenland ice stream were established two millennia ago. *Nature communications*, 15(1):1193.

Paterson, W. (1991). Why ice-age ice is sometimes "soft". *Cold Regions Science and Technology*, 20(1):75–98.

Peters, M. E., Blankenship, D. D., and Morse, D. L. (2005). Analysis techniques for coherent airborne radar sounding: Application to west antarctic ice streams. *Journal of Geophysical Research: Solid Earth*, 110(B6).

Scanlan, K. M., Buhl, D. P., and Blankenship, D. D. (2022). Polarimetric Airborne Radar Sounding as an Approach to Characterizing Subglacial Röthlisberger Channels. *IEEE Journal of Selected Topics in Applied Earth Observations and Remote Sensing*, 15:4455–4467.